# GTR-Loc: Geospatial Text Regularization Assisted Outdoor LiDAR Localization

**Shangshu Yu**[1], **Wen Li**[2,3], **Xiaotian Sun**[2,3], **Zhimin Yuan**[4],
**Xin Wang**[1], **Sijie Wang**[5], **Rui She**[6], **Cheng Wang**[2,3*]

[1]School of Computer Science and Engineering, Northeastern University, Shenyang 110819, China
[2]Fujian Key Laboratory of Sensing and Computing for Smart Cities, Xiamen University, China
[3]Key Laboratory of Multimedia Trusted Perception and Efficient Computing,
Ministry of Education of China, Xiamen University, China
[4]School of Artificial Intelligence and Software Engineering, Nanyang Normal University, China
[5]Nanyang Technological University, Singapore
[6]Beihang University, China
yushangshu@cse.neu.edu.cn     cwang@xmu.edu.cn

## Abstract

Prevailing scene coordinate regression methods for LiDAR localization suffer from localization ambiguities, as distinct locations can exhibit similar geometric signatures — a challenge that current geometry-based regression approaches have yet to solve. Recent vision–language models show that textual descriptions can enrich scene understanding, supplying potential localization cues missing from point cloud geometries. In this paper, we propose GTR-Loc, a novel text-assisted LiDAR localization framework that effectively generates and integrates geospatial text regularization to enhance localization accuracy. We propose two novel designs: a Geospatial Text Generator that produces discrete pose-aware text descriptions, and a LiDAR-Anchored Text Embedding Refinement module that dynamically constructs view-specific embeddings conditioned on current LiDAR features. The geospatial text embeddings act as regularization to effectively reduce localization ambiguities. Furthermore, we introduce a Modality Reduction Distillation strategy to transfer textual knowledge. It enables high-performance LiDAR-only localization during inference, without requiring runtime text generation. Extensive experiments on challenging large-scale outdoor datasets, including QEOxford, Oxford Radar RobotCar, and NCLT, demonstrate the effectiveness of GTR-Loc. Our method significantly outperforms state-of-the-art approaches, notably achieving a 9.64%/8.04% improvement in position/orientation accuracy on QEOxford. Our code is available at `https://github.com/PSYZ1234/GTR-Loc`.

## 1 Introduction

Accurate and robust LiDAR localization, which estimates the position and orientation of LiDAR sensors, is fundamental to autonomous vehicles and robotics. Traditional approaches [17, 5, 8, 44, 11] typically perform localization by matching a query point cloud to a pre-built 3D map. Although effective, these methods often incur high storage costs for 3D maps [52] and substantial communication overhead [22], limiting their wide applications in large-scale outdoor environments.

End-to-end regression models have recently propelled LiDAR localization forward, overcoming limitations of previous methods by enabling deep networks to learn scene-specific representations. These models mainly fall into two categories based on different regression objectives: Absolute Pose

---

*Corresponding author.

39th Conference on Neural Information Processing Systems (NeurIPS 2025).

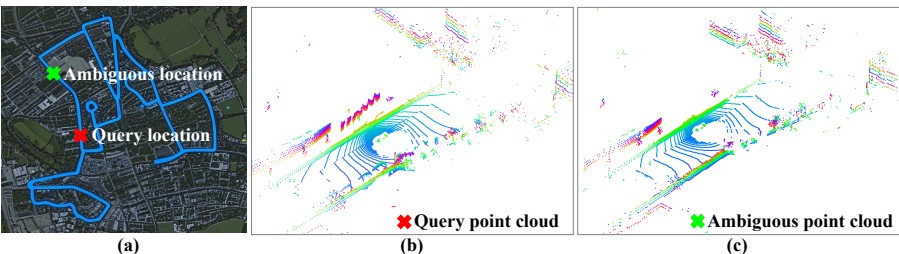

Figure 1: Illustration of LiDAR-localization ambiguities: panel (a) marks the true query location (red cross) and a spatially distinct ambiguous location (green cross), whereas panels (b) and (c) show the two sites' LiDAR point clouds, whose geometries appear nearly identical despite the distance.

Regression (APR) and Scene Coordinate Regression (SCR). APR models [41, 46, 48, 47, 40, 22, 12] directly regress the 6-DoF pose from point clouds, offering fast inference via compact architectures, yet often compromise accuracy due to limited geometric exploitation. Differently, SCR methods [23, 45, 21] predict world scene coordinates for each point, and then solve for the pose using RANSAC. SCR enforces geometric consistency during training, usually leading to improved performance.

However, current SCR methods suffer from a critical challenge—localization ambiguity arising from scene similarity, as shown in Fig. 1. This ambiguity occurs because distinct locations in large-scale outdoor environments often share highly similar local geometric structures. Consequently, methods (like SGLoc [23] and LightLoc [21]) relying only on local point cloud geometric features struggle to disambiguate visually similar, spatially distinct areas. This often results in erroneous coordinate predictions and subsequent localization failure. While LiSA [45] attempts to mitigate this ambiguity using semantic segmentation, geometrically similar regions often share similar semantic attributes, leaving the inherent problem fundamentally unsolved.

Recent advancements [1, 19, 13] in vision-language models highlight text's potent ability to enrich scene description and understanding. This insight motivates us to integrate textual cues that supply crucial scene localization information missing from common geometric data, thereby reducing localization ambiguities and improving performance. However, conventional text descriptions can be subjective, inconsistent across different times, or ambiguous for continuous observations [31, 50], making text-enhanced localization particularly challenging.

This paper proposes GTR-Loc, a novel text-assisted localization method that enhances SCR by generating and refining geospatial text regularization, thereby effectively mitigating ambiguities in LiDAR localization. Specifically, we propose two novel designs: Geospatial Text Generator (GTG) and LiDAR-Anchored Text Embedding Refinement (LATER). First, we propose a GTG to produce formatted text directly conveying discrete pose information. We partition geospatial positions and orientations into discrete districts and directions, then generate text based on the point cloud's current position and orientation. Unlike describing scene layout or the objects present, GTG directly constructs formatted text descriptions relevant to localization. Second, we propose LATER, a module that dynamically produces view-specific text embeddings. We learn a Transformer to leverage point cloud features to condition the instantiation of GTG, constructing refined LiDAR-text representations specific to the immediate view. By incorporating visual diversity within each district and direction category, it offers more effective regularization to reduce localization ambiguities. Finally, we introduce a Modality Reduction Distillation (MRD) strategy to enable LiDAR-only localization at inference, maintaining both effectiveness and efficiency. It distills textual regularization via a feature distillation module coupled with a distillation loss, transferring knowledge from LiDAR-text SCR to pure LiDAR SCR. Our contributions can be summarized as follows:

- We propose GTR-Loc, a novel text-assisted LiDAR localization framework. GTR-Loc is the first work to effectively design and integrate geospatial text descriptions as regularization to improve LiDAR SCR, leading to promising localization performance.
- We propose a Geospatial Text Generator and a LiDAR-Anchored Text Embedding Refinement module to dynamically create view-specific text descriptions focused on discrete pose information, providing significantly enhanced disambiguation capabilities for LiDAR localization.
- We devise a Modality Reduction Distillation strategy to enable LiDAR-only localization during inference. Extensive experiments on QEOxford [23], Oxford [2], and NCLT [26] datasets demon-

strate the great effectiveness of GTR-Loc, particularly outperforming state-of-the-art methods by 9.64%/8.04% on QEOxford.

## 2 Related Work

**LiDAR Localization**. Traditional LiDAR localization approaches typically aim to establish correspondences between the query point cloud and a pre-built 3D map. These approaches primarily achieve localization via map matching, such as PointNetVLAD [36], SOE-Net [43], SC$^2$-PCR++ [8], and TDM-RPMNet [49]. While potentially accurate, map-matching approaches suffer from expensive map storage, often at the terabyte to petabyte scale [52]. To address these limitations, regression-based localization is proposed, enabling inference without relying on pre-built 3D maps.

**Absolute Pose Regression (APR).** APR methods [15, 14, 32, 37, 33, 7, 34, 6] directly regress the 6-DoF pose from the input view (e.g., image or point cloud). Pioneering LiDAR APR works like PointLoc [41], employing PointNet++ [29] with an MLP head, demonstrate the approach's feasibility but show limitations in complex scenes. Subsequent works enhance APR with new architectures and loss functions. Notable examples include PosePN [46] with universal encoding and memory-aware regression; HypLiLoc [40] introducing cross-modal fusion and regression; and FlasMix [12] focusing on training acceleration. In the realm of multi-frame LiDAR localization, STCLoc [48] enforces spatio-temporal consistency across consecutive LiDAR scans, NIDALoc [47] draws inspiration from neurobiologically inspired mechanisms, and DiffLoc [22] refines poses through an iterative diffusion process. Nevertheless, APR's reliance on global scene representations can hinder the effective geometric encoding and potentially limit localization accuracy.

**Scene Coordinate Regression (SCR).** SCR methods [4, 39, 3, 38, 25] learns to predict per-point world coordinates, with RANSAC determining the final pose. SGLoc [23] first decouples localization into point correspondence regression and pose estimation. LiSA [45] distills semantic knowledge to enhance SCR. LightLoc [21] learns large-scale outdoor localization within 1 hour. Although SCR methods usually achieve higher localization accuracy than APR, they suffer from ambiguities arising from similar scenes, resulting in unreliable localization. To overcome the limitations of current methods, we propose GTR-Loc, a novel text-assisted LiDAR SCR framework. By incorporating and distilling geospatial textual regularization, we can effectively reduce localization ambiguities.

**Vision-Language Models.** Recent Large Language Models (LLMs), e.g., ChatGPT [27], LLaMA [35], and PaLM [9], have emerged as a promising way for understanding human languages. The success of LLMs has ignited interest in the Vision-Language (VL) research area. Foundational VL models like CLIP [31], COCOOP [51], and VL-Mamba [30] employ large-scale pre-training to align visual and textual representations.

These models also enable remarkable progress in diverse downstream tasks [1, 20, 19]. Since language and text can offer high-level scene descriptions, recent works explore their potential to address challenges in SLAM. LP-SLAM [50] and TextSLAM [18] first enable place recognition based on text labels within the SLAM system. Subsequently, Text2Pos [16], Text2Loc [42], and MNCL [24] propose using text to perform large-scale urban place recognition. Whereas existing approaches merely match text queries with images or point clouds, we embed structured text inside the pose estimation pipeline, enabling direct localization. We generate view-specific, pose-aware text descriptions as regularization for enhanced localization. Furthermore, we introduce a novel distillation strategy, enabling high-performance LiDAR-only localization during inference.

## 3 Method

### 3.1 Problem Formulation and Overview

**Problem Formulation.** The standard SCR framework for LiDAR-based localization aims to learn a mapping function, $f_\theta$, parameterized by $\theta$. Given an input LiDAR point cloud $P = \{p_i \in \mathbb{R}^3\}_{i=1}^N$, where each $p_i$ is the point in the sensor's local frame, the objective is to predict the corresponding world scene coordinates $\hat{C} = \{\hat{c}_i \in \mathbb{R}^3\}_{i=1}^N$ for each point $\hat{c}_i$, such that $\hat{C} = f_\theta(P)$. Subsequently, the 6-DoF pose $[t, q]$ (a translation vector $t \in \mathbb{R}^3$ and a rotation vector $q \in \mathbb{R}^3$) can be estimated from the predicted point-to-point correspondences, typically using a RANSAC-based algorithm. In this paper, we formulate the SCR learning as multimodal regression, incorporating LiDAR point clouds

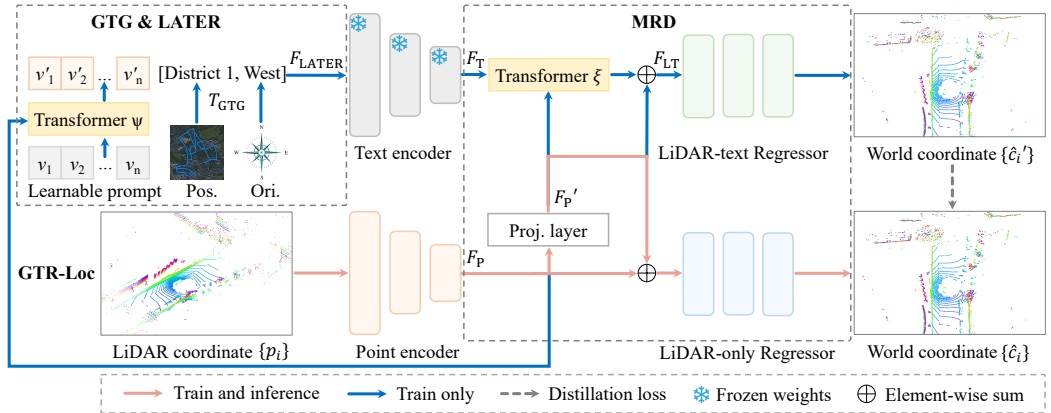

Figure 2: Overview of our method. GTR-Loc enhances LiDAR localization with text assistance to regularize SCR: a Geospatial Text Generator (**GTG**) provides discrete pose-aware text descriptions $T_{\text{GTG}}$, and a LiDAR-Anchored Text Embedding Refinement (**LATER**) module dynamically constructs view-specific text embeddings $F_{\text{LATER}}$ conditioned on point features $F_{\text{P}}$. A Transformer $\xi$ is employed to fuse multimodal features for LiDAR-text regression. Furthermore, a Modality Reduction Distillation (**MRD**) strategy enables LiDAR-only inference by distilling textural regularization.

with an additional input, geospatial text description $T$. Then, we learn a new mapping function $g_\phi$, parameterized by $\phi$, that uses both inputs to perform point regression, depicted as $\hat{C} = g_\phi(P, T)$.

**Overview.** This paper proposes GTR-Loc, a novel text-assisted LiDAR SCR framework, to address the challenge of ambiguities in LiDAR localization. Fig. 2 illustrates the network architecture of GTR-Loc. First, the point encoder extracts point cloud features $F_{\text{P}}$ from the input LiDAR scan. Concurrently, we create discrete pose-aware text descriptions for each view to aid localization. Specifically, we propose two novel designs: Geospatial Text Generator (GTG) and LiDAR-Anchored Text Embedding Refinement (LATER) module. The proposed GTG (Sec. 3.2) generates formatted text descriptions $T_{\text{GTG}}$ that directly convey discrete position and orientation information, which is relevant to localization. Then, the proposed LATER (Sec. 3.3) dynamically produces view-specific text embeddings $F_{\text{LATER}}$ by incorporating current $F_{\text{P}}$, regularizing the SCR network for reducing ambiguities. Finally, a Transformer $\xi$ combines text embeddings $F_{\text{T}}$ with point cloud features $F_{\text{P}}^{'}$ and feeds the result $F_{\text{LT}}$ to the LiDAR-text regressor. Furthermore, we propose Modality Reduction Distillation (Sec. 3.4) to distill geospatial text regularization, enabling LiDAR-only localization during inference. To supervise the model training, we adopt standard L1 losses (Sec. 3.5) to constrain the predicted and ground truth points. The SCR framework mostly follows the architecture of LightLoc [21], which consists of a multi-scale encoder and a regressor. We encode text using the text encoder of a pre-trained CLIP model [31]. Detailed descriptions are provided below.

## 3.2 Geospatial Text Generator

Traditional Vision-Language models [50, 18] leveraging text for SLAM often focus on generating descriptions of surrounding environments, e.g., scene layout or prominent objects. For example, a description of the scene point cloud is illustrated in Fig. 3 (b). While intuitive, relying on such descriptive text for LiDAR localization may face significant challenges. When revisiting the same location, dynamic changes in the scene can result in inconsistent textual descriptions. This inconsistency can significantly impede robust localization. In addition, describing scene content provides only indirect cues for localization, falling short of the direct position and orientation information required for 6-DoF pose estimation. To overcome these limitations, we propose a Geospatial Text Generator (GTG) to produce formatted pose-aware text descriptions that directly aid LiDAR localization.

Specifically, GTG generates concise descriptions grounded in predefined position and orientation partitions, as shown in Fig. 3 (c). We first divide the planar map into $M$ distinct geographical districts to describe the LiDAR sensor's position. We define a mapping function $\mathcal{M} : x \to \{1, ..., M\}$, which maps a planar position $x \in \mathbb{R}^2$ of the input point cloud $P$ to a discrete district identifier $z = \mathcal{M}(x)$. Concurrently, we discretize the compass directions into $K$ directional bins to indicate the LiDAR

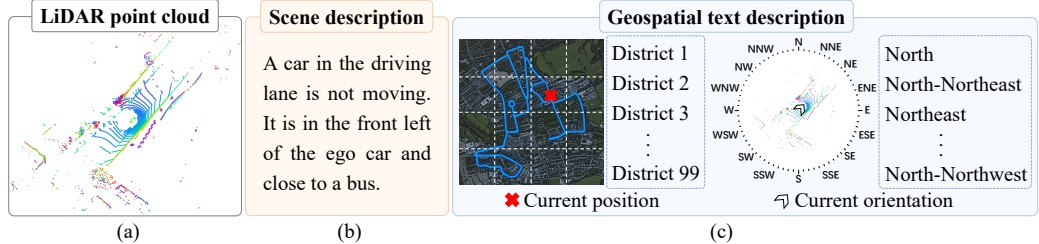

Figure 3: Scene description comparison. (a) Input LiDAR point cloud. (b) A free-form scene description, which is often subjective and verbose. (c) Our structured geospatial text description provides discrete pose cues for localization by encoding a district ID and a quantized direction.

sensor's heading. The direction set $\mathcal{D}$ includes cardinal (e.g., North), intercardinal (e.g., Northeast), and finer-grained bearings (e.g., North-Northeast) to capture more precise orientation. The orientation angle $\theta \in [0, 2\pi)$ of the input point cloud $P$ is then mapped to a discrete direction bin $d$ via a discretization function $\mathcal{O}$, such that $d = \mathcal{O}(\theta), d \in \mathcal{D}$. Finally, an example of the $T_{\text{GTG}}$ template can be expressed as "District 99, West-Southwest."

These discretely determined identifiers, district $z$ and direction $d$, represent the current geospatial state. Compared to conventional scene layout or object-centric descriptions, the proposed GTG provides more stable and informative cues for localization. For example, the constructed text, e.g., "District 99, West-Southwest", yields a signal that is stable and repeatable whether the scene changes. Conversely, free-form descriptions of the scene are subject to change due to moving objects and construction. Hence, the discrete pose-aware text in a standardized format offers clear advantages in aiding localization and solving ambiguities. The generated descriptions $T_{\text{GTG}}$ are then input to the LATER module, which builds upon them to construct refined textual representations crucial for reducing scene ambiguities and thereby improving LiDAR localization performance.

### 3.3 LiDAR-Anchored Text Embedding Refinement

While GTG provides standardized, pose-aware text descriptions, its static prompts usually fail to capture the visual diversity within each district and direction category. For instance, consecutive LiDAR scans within the same district or direction may differ geometrically due to viewpoint shifts, occlusions, or environmental variations [16, 42, 24]. However, $T_{\text{GTG}}$ yields identical geospatial text for these LiDAR point clouds. This consequently obscures fine-grained differences, leading to descriptive ambiguities that can impede high-accuracy localization. To address this, we propose a LiDAR-Anchored Text Embedding Refinement (LATER) module to refine $T_{\text{GTG}}$ based on input point cloud features. Then, we can ensure that the refined text embeddings dynamically adapt to the unique characteristics of every input point cloud.

To be specific, we learn a Transformer decoder, parameterized by $\psi$, to generate conditional text embeddings for each LiDAR point cloud. As shown in Fig. 2, the Transformer takes two primary sets of inputs: $n$ learnable prompts $\{v_1, v_2, ..., v_n\}$ and view-specific point cloud features $F_{\text{P}}$. The prompts are represented as learnable parameters using `nn.Parameter`. The feature $F_{\text{P}}$ is directly extracted from the input point cloud. Within the Transformer, both self-attention and cross-attention mechanisms are utilized layer-wise to fuse these modalities. Self-attention stays inside $v_n$; cross-attention then connects $v_n$ (as queries) to $F_{\text{P}}$ (as keys/values). Finally, the refined text embedding $F_{\text{LATER}}$ is constructed by combining the Transformer $\psi$'s output $v_n'$ with $W_{\text{GTG}}$, depicted as:

$$\{v_1', v_2', ..., v_n'\} = Transformer_\psi(Q = \{v_1, v_2, ..., v_n\}, K, V = F_{\text{P}}), \qquad (1)$$

$$F_{\text{LATER}} = \{v_1', v_2', ..., v_n', W_{\text{GTG}}\}, \qquad (2)$$

where $W_{\text{GTG}}$ is the word embedding of $T_{\text{GTG}}$.

The LiDAR anchoring mechanism refines the initial static prompt $T_{\text{GTG}}$ with visual evidence from the input point cloud, effectively yielding view-specific textual embeddings $F_{\text{LATER}}$. It provides a more discriminative representation to distinguish subtle variations even within the same district or direction category. $F_{\text{LATER}}$ not only aligns and adapts more closely to the input LiDAR scan, but

also delivers critical pose information for localization. Previous VLMs bridge vision and language via methods like COCOOP's learnable prompts [51], BLIP's contrastive-fusion learning [20], or Flamingo's gated cross-attention for injecting visual tokens into frozen LLMs [1]. However, they are designed for image understanding tasks such as image captioning or zero-shot classification, while LATER is specifically designed for LiDAR localization. Hence, it can effectively regularize SCR to reduce localization ambiguities. $F_{\text{LATER}}$ is then processed by a frozen CLIP text encoder to produce the text embedding $F_{\text{T}}$ for SCR.

## 3.4 Modality Reduction Distillation

Although the proposed geospatial text is effective, its construction relies on ground truth 6-DoF poses. This dependency is impractical for inference, as the pose is the very thing our localization task aims to estimate. Meanwhile, the absence of text or the presence of inaccurate text can lead to erroneous localization results. Hence, we introduce a Modality Reduction Distillation (MRD) strategy designed to transfer the knowledge from LiDAR-text SCR to pure LiDAR SCR. This allows us to achieve high-performance localization with only LiDAR inputs at inference, while retaining regularization imposed by the geospatial text.

As shown in Fig. 2, the multimodal regressor (LiDAR-text) acts as the teacher, while the pure point cloud regressor (LiDAR-only) acts as the student. We introduce a projection layer, a 3-layer MLP with identical hidden size, to transform point cloud features $F_{\text{P}}$ into $F_{\text{P}}^{'}$. The text embedding $F_{\text{T}}$ and $F_{\text{P}}^{'}$ are then integrated into a new Transformer decoder, parameterized by $\xi$, to be aggregated for the subsequent regression. The fused LiDAR-text features $F_{\text{LT}}$ for SCR are depicted as:

$$F_{\text{LT}} = F_{\text{P}}^{'} + \alpha Transformer_{\xi}(Q = F_{\text{T}}, K, V = F_{\text{P}}^{'}). \tag{3}$$

Hence, the projection layer serves a dual role: it enables multimodal feature fusion for LiDAR-text SCR and, through a skip connection with $F_{\text{P}}$, contributes to LiDAR-only SCR as well. This allows for simultaneous feature enhancement and knowledge distillation. Finally, we define a distillation loss $\mathcal{L}_{\text{D}}$ to measure the discrepancy between the output of the teacher and student, represented as:

$$\mathcal{L}_{\text{D}} = \sum_{i=1}^{N} \|\hat{c}_i - \hat{c}_i^{'}\|_1, \tag{4}$$

where $\hat{c}_i$ and $\hat{c}_i^{'}$ denote the predicted world coordinates from the LiDAR-only regressor and LiDAR-text regressor, respectively. MRD enables the student regressor to mimic the teacher's refined outputs. The student thus inherits the text-assisted disambiguation, learning to narrow the search space using only its point cloud input. Once training is complete, the student SCR delivers accurate LiDAR-only localization at inference—no runtime text generation required.

## 3.5 Loss Functions

Our GTR-Loc architecture employs a multi-head learning framework that jointly optimizes LiDAR-text SCR and LiDAR-only SCR. The main optimization objective is to learn dense point predictions for localization. Hence, the proposed model is supervised by two loss functions during training:

$$\mathcal{L}_{\text{LT}} = \sum_{i=1}^{N} \|\hat{c}_i^{'} - c_i\|_1, \mathcal{L}_{\text{LO}} = \sum_{i=1}^{N} \|\hat{c}_i - c_i\|_1, \tag{5}$$

where $c_i$ is the ground trurh world coordinates. The overall loss function $\mathcal{L}_{\text{SCR}}$ is a weighted summation of $\mathcal{L}_{\text{LT}}$, $\mathcal{L}_{\text{LO}}$, and the distillation loss $\mathcal{L}_{\text{D}}$ with a balancing weight $\beta$, represented as:

$$\mathcal{L}_{\text{SCR}} = \beta_1 \mathcal{L}_{\text{LT}} + \beta_2 \mathcal{L}_{\text{LO}} + \beta_3 \mathcal{L}_{\text{D}}. \tag{6}$$

# 4 Experiments

## 4.1 Experiment Settings

**Datasets.** We evaluate GTR-Loc on three commonly used large-scale outdoor datasets: Oxford Radar RobotCar (Oxford) [2], QEOxford [23], and NCLT [26]. **Oxford** is a large-scale urban dataset

Table 1: Quantitative results on the QEOxford dataset. We report the position error (m) and orientation error (°). Mechanism (Mech.) types are denoted as follows: MA for Multi-frame APR; SA for Single-frame APR; and SS for Single-frame SCR. Test Frames (TFs) denote the number of point cloud frames used during testing. We highlight the **best** and second-best results.

| Methods | Mech. | TFs | 15-13-06-37 | 17-13-26-39 | 17-14-03-00 | 18-14-14-42 | Avg. [m/°] |
|---|---|---|---|---|---|---|---|
| STCLoc [48] | MA | 3 | 5.14/1.27 | 6.12/1.21 | 5.32/1.08 | 4.76/1.19 | 5.34/1.19 |
| NIDALoc [47] | MA | 5 | 3.71/1.50 | 5.40/1.40 | 3.94/1.30 | 4.08/1.30 | 4.28/1.38 |
| DiffLoc [22] | MA | 3 | **2.03/1.04** | **1.78/0.79** | **2.05/0.83** | **1.56/0.83** | **1.86/0.87** |
| PointLoc [41] | SA | 1 | 10.75/2.36 | 11.07/2.21 | 11.53/1.92 | 9.82/2.07 | 10.79/2.14 |
| PosePN [46] | SA | 1 | 9.47/2.80 | 12.98/2.35 | 8.64/2.19 | 6.26/1.64 | 9.34/2.25 |
| PosePN++ [46] | SA | 1 | 4.54/1.83 | 6.44/1.78 | 4.89/1.55 | 4.64/1.61 | 5.13/1.69 |
| PoseMinkLoc [46] | SA | 1 | 6.77/1.84 | 8.84/1.84 | 8.08/1.69 | 6.56/2.06 | 7.56/1.86 |
| PoseSOE [46] | SA | 1 | 4.17/1.76 | 6.16/1.81 | 5.42/1.87 | 4.16/1.70 | 4.98/1.79 |
| HypLiLoc [40] | SA | 1 | 5.03/1.46 | 4.31/1.43 | 3.61/1.11 | 2.61/1.09 | 3.89/1.27 |
| FlashMix [12] | SA | 1 | 2.04/1.95 | 1.95/1.83 | 2.44/2.18 | 2.81/2.14 | 2.31/2.03 |
| SGLoc [23] | SS | 1 | 1.79/1.67 | 1.81/1.76 | 1.33/1.59 | 1.19/1.39 | 1.53/1.60 |
| LiSA [45] | SS | 1 | 0.94/1.10 | 1.17/1.21 | 0.84/1.15 | 0.85/1.11 | 0.95/1.14 |
| LightLoc [21] | SS | 1 | 0.82/1.12 | 0.85/1.07 | 0.81/1.11 | 0.82/1.16 | 0.83/1.12 |
| **GTR-Loc** | SS | 1 | **0.77/1.02** | **0.77/1.01** | **0.67/1.01** | **0.80/1.07** | **0.75/1.03** |

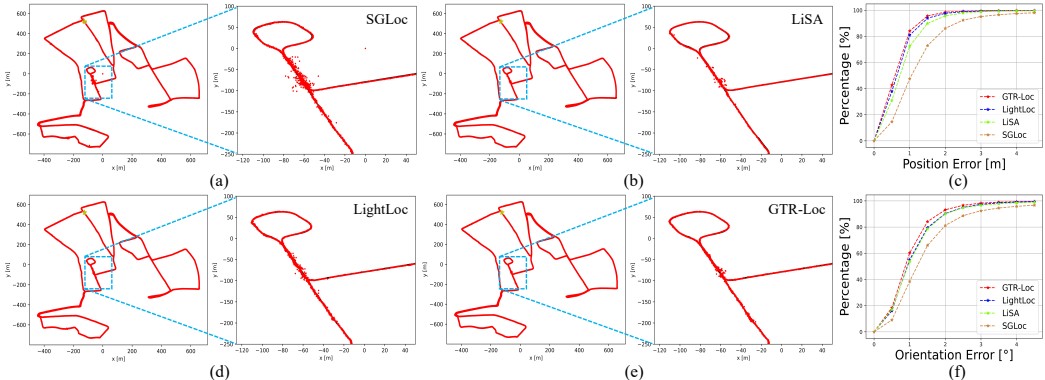

Figure 4: Visual comparisons on QEOxford. (a) (b) (d) (e): predicted trajectories (red) overlaid on ground truth (black); a star marks the starting position, and the blue box highlights a challenging road segment. (c) (f): cumulative error distribution curves for position (top) and orientation (bottom).

collected using a Nissan LEAF platform equipped with dual Velodyne HDL-32E sensors, providing dense LiDAR scans across multiple 10km trajectories within a 2km² city area. This dataset captures diverse variations in weather and traffic density. **QEOxford** is a quality-enhanced version of the Oxford dataset, more suitable for localization evaluation. The original Oxford dataset is further refined to mitigate the inherent errors in raw GPS/INS measurements. **NCLT** is a campus-scale dataset collected by a Segway robot with a Velodyne HDL-32E, covering 5.5km of trajectories within a 0.45km² area. This dataset encompasses both complex outdoor and indoor environments.

**Baselines and Evaluation Metrics.** To evaluate localization performance, we compare our approach against various LiDAR-based SCR and APR methods. For SCR, we include SGLoc [23], LiSA [45] and LightLoc [21], the latter being the state-of-the-art (SOTA) method. For APR, we evaluate against single-frame methods including PointLoc [41], PosePN [46], PosePN++[46], PoseSOE[46], PoseMinkLoc [46], HyLiLoc [40], and FlashMix [12], as well as multi-frame methods such as STCLoc [48], NIDALoc [47], and DiffLoc [22]. Following previous methods [22, 21], we report the position/orientation error [m/°] for each trajectory and the overall average across all trajectories.

**Implementation Details.** In this paper, we employ LightLoc [21] as the SCR backbone and a pre-trained CLIP [31] as the text encoder. Following LightLoc, we load the pre-trained encoder weights. We adopt the AdamW optimizer with a one-cycle learning rate schedule ranging from $5e-4$ to $5e-3$. The model is trained for 25 epochs on Oxford/QEOxford and 30 epochs on NCLT. Input

Table 2: Quantitative results on the Oxford dataset. The notations follow Tab. 1.

| Methods | Mech. | TFs | 15-13-06-37 | 17-13-26-39 | 17-14-03-00 | 18-14-14-42 | Avg. [m/°] |
|---|---|---|---|---|---|---|---|
| STCLoc [48] | MA | 3 | 6.93/1.48 | 7.55/1.23 | 7.44/1.24 | 6.13/1.15 | 7.01/1.28 |
| NIDALoc [47] | MA | 5 | 5.45/1.40 | 7.63/1.56 | 6.68/1.26 | 4.80/1.18 | 6.14/1.35 |
| DiffLoc [22] | MA | 3 | **3.57/0.88** | **3.65/0.68** | **4.03/0.70** | **2.86/0.60** | **3.53/0.72** |
| PointLoc [41] | SA | 1 | 12.42/2.26 | 13.14/2.50 | 12.91/1.92 | 11.31/1.98 | 12.45/2.17 |
| PosePN [46] | SA | 1 | 14.32/3.06 | 16.97/2.49 | 13.48/2.60 | 9.14/1.78 | 13.48/2.48 |
| PosePN++ [46] | SA | 1 | 9.59/1.92 | 10.66/1.92 | 9.01/1.51 | 8.44/1.71 | 9.43/1.77 |
| PoseMinkLoc [46] | SA | 1 | 11.20/2.62 | 14.24/2.42 | 12.35/2.46 | 10.06/2.15 | 11.96/2.41 |
| PoseSOE [46] | SA | 1 | 7.59/1.94 | 10.39/2.08 | 9.21/2.12 | 7.27/1.87 | 8.62/2.00 |
| HypLiLoc [40] | SA | 1 | 6.88/**1.09** | 6.79/1.29 | 5.82/**0.97** | 3.45/**0.84** | 5.74/**1.05** |
| FlashMix [12] | SA | 1 | 3.05/1.96 | 4.55/2.05 | 4.67/2.05 | 2.94/1.79 | 3.80/1.96 |
| SGLoc [23] | SS | 1 | 3.01/1.91 | 4.07/2.07 | 3.37/1.89 | 2.12/1.66 | 3.14/1.88 |
| LiSA [45] | SS | 1 | 2.36/1.29 | 3.47/1.43 | 3.19/1.34 | 1.95/1.23 | 2.74/1.32 |
| LightLoc [21] | SS | 1 | 2.33/1.21 | 3.19/1.34 | 3.11/1.24 | 2.05/1.20 | 2.67/1.25 |
| **GTR-Loc** | SS | 1 | **2.29**/1.17 | **3.07/1.21** | **2.99**/1.20 | **2.00**/1.18 | **2.59**/1.19 |

point clouds are voxel-downsampled with a voxel size of 0.25m on Oxford/QEOxford and 0.3m on NCLT. The number of districts $z$ is set to 100, and the number of directions $d$ is set to 16. $\alpha$ in Eq. 3 is set to 0.1. $\beta_1$ and $\beta_2$ in Eq. 6 are set to 1, $\beta_3$ is set to 0.1. GTR-Loc is implemented in PyTorch [28] and MinkowskiEngine [10]. All experiments are conducted on a single NVIDIA RTX 4090 GPU.

## 4.2 Comparison With State-of-the-Art Methods

**Results on Oxford.** We first evaluate GTR-Loc on the challenging QEOxford dataset. As shown in Tab. 1, GTR-Loc achieves SOTA accuracy among all single-frame SCR and APR methods, with a mean position/orientation error of 0.75m/1.03°. This performance significantly surpasses the previous leading SCR method, LightLoc (0.83m/1.12°), by 9.64%/8.04%, showcasing our effectiveness. Although our method processes only a single frame, it still surpasses DiffLoc, the SOTA multi-frame APR approach, by a substantial margin in positional accuracy. The results shown in Tab. 2 further demonstrate the superiority of our method (2.59m/1.19°) on the original Oxford dataset. This notable improvement highlights the effectiveness of integrating geospatial text regularization in resolving localization ambiguities, particularly in complex urban environments like Oxford.

To provide further insights into performance, we visualize representative predicted trajectories and cumulative error distribution curves on the QEOxford dataset. As illustrated in Fig. 4, our estimated trajectory on sequence 17-14-03-00 closely follows the ground truth trajectory over the entire sequence. As marked by blue boxes, GTR-Loc maintains stable and accurate tracking while competitor trajectories display systematic offsets, highlighting its robustness. The cumulative error distribution curves offer a quantitative summary supporting these observations. The sharper rise of our curves indicates consistently higher accuracy across most trajectory points. For example, over 85% of GTR-Loc's position errors fall below 1 m, compared to just 80% for LightLoc.

**Results on NCLT.** The NCLT dataset involves long-term data collection in a diverse setting, spanning both outdoor and indoor campus environments. As presented in Tab. 3, GTR-Loc achieves leading performance on position accuracy. Our approach yields mean position/orientation errors of 1.40m/2.62°, ranking first and second in position and orientation among single-frame SCR and APR methods. Our approach also delivers performance comparable to that of SOTA multi-frame APR methods, yet it operates using only a single frame during inference, thereby offering greater flexibility. This dataset presents unique difficulties like abrupt environmental transitions and varying structural complexity. GTR-Loc incorporates geospatial cues that remain informative across indoor and outdoor settings, resolving ambiguities in perceptually similar areas.

**Runtime Analysis.** We report the training (h)/inference (ms) time in Tab. 4. The entire model training process lasted approximately 4 hours, with a peak GPU memory consumption of around 10 GB. On the Oxford/QEOxford datasets, the average inference time per sample is 29ms (34 FPS), and on the NCLT dataset, it is 48ms (21 FPS). These speeds are well within the respective scanning frequencies of 20 Hz for Oxford/QEOxford and 10 Hz for NCLT, highlighting the model's ability to maintain high accuracy during real-time operation.

Table 3: Quantitative results on the NCLT dataset. The notations follow Tab. 1.

| Methods | Mech. | TFs | 2012-02-12 | 2012-02-19 | 2012-03-31 | 2012-05-26 | Avg. [m/°] |
|---|---|---|---|---|---|---|---|
| STCLoc [48] | MA | 3 | 4.91/4.34 | 3.25/3.10 | 3.75/4.04 | 7.53/4.95 | 4.86/4.11 |
| NIDALoc [47] | MA | 5 | 4.48/3.59 | 3.14/2.52 | 3.67/3.46 | 6.32/4.67 | 4.40/3.56 |
| DiffLoc [22] | MA | 3 | **0.99/2.40** | **0.92/2.14** | **0.98/2.27** | **1.36/2.48** | **1.06/2.32** |
| PointLoc [41] | SA | 1 | 7.23/4.88 | 6.31/3.89 | 6.71/4.32 | 9.55/5.21 | 7.45/4.58 |
| PosePN [46] | SA | 1 | 9.45/7.47 | 6.15/5.05 | 5.79/5.28 | 12.32/7.42 | 8.43/6.31 |
| PosePN++ [46] | SA | 1 | 4.97/3.75 | 3.68/2.65 | 4.35/3.38 | 8.42/4.30 | 5.36/3.52 |
| PoseMinkLoc [46] | SA | 1 | 6.24/5.03 | 4.87/3.94 | 4.23/4.03 | 9.32/6.11 | 6.17/4.78 |
| PoseSOE [46] | SA | 1 | 13.09/8.05 | 6.16/4.51 | 5.24/4.56 | 13.27/7.85 | 9.44/6.24 |
| HypLiLoc [40] | SA | 1 | 1.71/3.56 | 1.68/2.69 | 1.52/2.90 | **2.29**/3.34 | 1.80/3.12 |
| FlashMix [12] | SA | 1 | 2.59/4.27 | 1.54/3.26 | 1.42/3.65 | 4.96/5.80 | 2.63/4.25 |
| SGLoc [23] | SS | 1 | 1.20/3.08 | 1.20/3.05 | 1.12/3.28 | 3.48/4.43 | 1.75/3.46 |
| LiSA [45] | SS | 1 | 0.97/**2.23** | 0.91/**2.09** | 0.87/**2.21** | 3.11/**2.72** | 1.47/**2.31** |
| LightLoc [21] | SS | 1 | 0.98/2.76 | 0.89/2.51 | 0.86/2.67 | 3.10/3.26 | 1.46/2.80 |
| **GTR-Loc** | SS | 1 | **0.95**/2.53 | **0.82**/2.45 | **0.82**/2.52 | 3.01/2.99 | **1.40**/2.62 |

## 4.3 Ablation Study

**Effects of Geospatial Text Generator.** To comprehensively evaluate GTR-Loc, we first conduct ablation studies on the proposed Geospatial Text Generator (GTG). We report the average error for each dataset. As shown in Tab. 5, removing all proposed modules degrades performance from 0.75m/1.03° to 0.83m/1.12° (the vanilla model's result), highlighting the importance of geospatial text regularization. Then, we conduct experiments using different types of text descriptions. Specifically, we replace our geospatial text with a SOTA 3D scene description generator, TOD3Cap [13], that produces text describing scene layout and objects present. An example of the generated text for a scene is shown in Fig. 3 (b). Demonstrating no significant gains compared to the vanilla model on different datasets, this variant also remains markedly inferior to our full model. Human-like, free-form scene descriptions are inherently unstable for localization, as dynamic objects and environmental changes create inconsistent cues. In addition, they provide only indirect and noisy cues for pose estimation. Hence, the discrete pose-aware text generated by GTG delivers a more direct and effective localization cue than unconstrained scene descriptions, leading to better performance.

In addition, we conduct ablation studies by replacing the CLIP text encoder with embedding layers (ELs) to demonstrate the necessity of using textual representations. Specifically, we learn two separate, learnable embedding layers (`nn.Embedding`) initialized from scratch: one for the 100 district IDs and another for the 16 direction IDs. The resulting vectors are concatenated and passed through an MLP to match the feature dimension of the original CLIP embedding. The results shown in the table indicate that using simple learnable embeddings only leads to a small performance improvement. By processing geospatial text with CLIP, we can leverage pre-trained semantic priors, eliminating the need to learn complex semantic and spatial relationships from scratch.

**Effects of District $z$ and Direction $d$.** To investigate the impact of spatial partitioning granularity in GTG, we conduct an ablation study by varying the number of districts $z$ and directions $d$. We train and evaluate GTR-Loc with different configurations, varying $z$ across values like 49, 100, 144, and $d$ across values like 8, 16, 32. As shown in Tab. 6, localization accuracy improves as $z$ increases from 49 to 100 and $d$ from 8 to 16, indicating that finer geospatial discretization offers more discriminative contextual cues. However, further increasing $z$ and $d$ yields negligible gains, as excessive granularity fails to provide additional benefits in resolving localization ambiguities. The chosen values of $z$=100 and $d$=16 in this paper are appropriate for our method. This suggests that while sufficient granularity is needed for disambiguation, excessive partitioning may hinder performance.

Our method's partitioning scheme is both robust and scalable, making it practical for real-world, large-scale deployment. Its robustness stems from using an abstract coordinate grid rather than semantic environmental features, ensuring stability against long-term environmental changes like construction or seasonal variations. Results on the NCLT dataset, known for its data collection spanning several months, also demonstrate this. Furthermore, this design is also inherently scalable. The 10x10 uniform grid is illustrative and can be seamlessly extended to city-scale applications using

Table 4: Training (h)/Inference (ms) time.

| Method | QEOxford | Oxford | NCLT |
|--------|----------|--------|------|
| SGLoc | 50/38 | 50/38 | 42/75 |
| LiSA | 53/38 | 53/38 | 44/75 |
| LightLoc | 1/29 | 1/29 | 1/48 |
| **GTR-Loc** | 4/29 | 4/29 | 3/48 |

Table 5: Ablation of GTG.

| Method | QEOxford | Oxford | NCLT |
|--------|----------|--------|------|
| vanilla | 0.83/1.12 | 2.67/1.25 | 1.46/2.80 |
| TOD3Cap | 0.84/1.15 | 2.66/1.24 | 1.47/2.78 |
| ELs | 0.80/1.10 | 2.64/1.23 | 1.44/2.75 |
| **GTG** | 0.75/1.03 | 2.59/1.19 | 1.40/2.62 |

Table 6: Ablation of $z$ and $d$.

| Method | QEOxford | Oxford | NCLT |
|--------|----------|--------|------|
| $z$=49, $d$=8 | 0.79/1.09 | 2.62/1.21 | 1.43/2.67 |
| $z$=**100**, $d$=**16** | 0.75/1.03 | 2.59/1.19 | 1.40/2.62 |
| $z$=144, $d$=32 | 0.76/1.02 | 2.60/1.22 | 1.41/2.61 |

Table 7: Ablation of LATER.

| Method | QEOxford | Oxford | NCLT |
|--------|----------|--------|------|
| w/o LATER | 0.78/1.10 | 2.62/1.22 | 1.43/2.70 |
| Sum | 0.76/1.06 | 2.62/1.21 | 1.41/2.66 |
| Concatenate | 0.77/1.09 | 2.61/1.22 | 1.41/2.68 |
| **Transformer** | 0.75/1.03 | 2.59/1.19 | 1.40/2.62 |

Table 8: Ablation of MRD.

| Method | QEOxford | Oxford | NCLT |
|--------|----------|--------|------|
| **MRD (w/o text)** | 0.75/1.03 | 2.59/1.19 | 1.40/2.62 |
| MRD (w text) | 0.74/0.93 | 2.58/1.07 | 1.12/2.40 |

standard hierarchical grids (e.g., UTM). Our ablation study confirms its robustness as the number of partitions changes (e.g., from $z$=100 to $z$=49 or $z$=144).

**Effects of LiDAR-Anchored Text Embedding Refinement.** To evaluate the contribution of dynamically refining text embeddings using visual context, we conduct an ablation study on the proposed LiDAR-Anchored Text Embedding Refinement (LATER) module. We compare four distinct configurations: (1) a baseline using only the static prompt generated from GTG without LATER; (2) a simplified variant that replaces Transformer fusion with element-wise summation (3) another variant with feature concatenation fusion; and (4) our proposed LATER, which utilizes a Transformer to fuse text embeddings with point cloud features. The results reported in Tab. 7 clearly favor our proposed approach. Using only static prompts without LATER led to worse results, e.g., 0.78m/1.10° on QEOxford. The summation and feature concatenation fusion variant performs intermediately. The Transformer-based LATER achieves the best performance. It allows creating highly discriminative, scene-specific text embeddings using visual context for accurate localization.

**Effects of Modality Reduction Distillation.** We further evaluate the Modality Reduction Distillation (MRD) strategy in Tab. 8. We present ablation studies with text inputs, generated by ground truth poses, at inference. Results indicate that using text during inference usually leads to better performance, e.g., 0.74m/0.93° on QEOxford. However, generating text at inference can be impractical and introduce unwanted dependencies, since it depends on discrete pose estimates. The distilled model without text achieves performance nearly matching that of the text-assisted localization approach, e.g., 0.75m/1.03° on QEOxford. This demonstrates that MRD distills geospatial text regularization from the teacher effectively, enabling high-performance LiDAR-only localization during inference. In the Appendix, we provide additional ablation studies and visualizations for the Oxford and NCLT datasets, along with a discussion of limitations and future work.

## 5 Conclusion

This paper introduces GTR-Loc, the first text-assisted LiDAR localization framework that integrates geospatial text regularization into an SCR network to reduce localization ambiguities. The proposed geospatial text regularization consists of two components: a Geospatial Text Generator, which produces formatted, discrete pose-aware text descriptions, and a LiDAR-Anchored Text Embedding Refinement module, which dynamically constructs view-specific embeddings conditioned on current point cloud features. Furthermore, we introduce a Modality Reduction Distillation strategy to distill textual regularization knowledge, enabling high-performance LiDAR-only localization at inference time. Comprehensive experiments on challenging large-scale outdoor datasets, QEOxford, Oxford Radar Robotcar, and NCLT, demonstrate the effectiveness of GTR-Loc.

## Acknowledgments

This work was supported by the Fundamental Research Funds for the Central Universities under Grant N25XQD053. We would like to thank the anonymous reviewers for their valuable suggestions.

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

# Appendix

## A    More Experimental Results

**More Dataset Details.** Regarding the datasets, ground truth poses for the Oxford [1] and QEOxford [3] datasets are obtained through interpolation from an integrated GPS/INS system. For the NCLT [4] dataset, ground truth poses are generated post-collection using SLAM. Further details on the data splits can be found in Tab. 1 and Tab. 2.

**More Ablation Study.** Importantly, our deployed model (a distilled, LiDAR-only GTR-Loc) requires no text during inference and uses only a single LiDAR scan. However, we further investigate the case where text is used during inference. First, we train an auxiliary classification network to predict district and direction categories at test time. This network is designed with a LightLoc [2] encoder, followed by four fully-connected layers of identical feature dimensions, to separately predict partitions for position and orientation. The training configurations are consistent with those used for GTR-Loc. During inference, this classification network first predicts districts $z$ and directions $d$. This prediction is then used to generate a conditioned textual input for our primary localization network. The detailed localization results alongside the position/orientation classification accuracy of this auxiliary network are presented in Tab. 3 and Tab. 4. The results (row 2) indicate that this approach delivers virtually no performance gains, while it increases both test time and computational complexity owing to the additional classification network and the LiDAR-text regression. The limited classification accuracy in the specific dataset, i.e., NCLT, produces faulty text outputs, which in turn propagate errors into the localization results. We then test a hypothetical scenario where ground truth poses are available for text generation at inference. This better result (row 3) establishes the theoretical upper bound for performance achievable with text. However, it is impractical, as gt poses are unavailable during real-world inference. Therefore, using MRD distillation is a more viable solution.

Table 1: Details of the Oxford dataset.

| Sequence | Length (km) | Weather | Split |
|---|---|---|---|
| 11-14-02-26 | 9.37 | sunny | Train |
| 14-12-05-52 | 9.22 | overcast | Train |
| 14-14-48-55 | 9.05 | overcast | Train |
| 18-15-20-12 | 9.04 | overcast | Train |
| 15-13-06-37 | 8.85 | overcast | Eval |
| 17-13-26-39 | 9.02 | sunny | Eval |
| 17-14-03-00 | 9.02 | sunny | Eval |
| 18-14-14-42 | 9.04 | overcast | Eval |

Table 2: Details of the NCLT dataset.

| Sequence | Length (km) | Weather | Split |
|---|---|---|---|
| 2012-01-22 | 6.10 | overcast | Train |
| 2012-02-02 | 6.20 | sunny | Train |
| 2012-02-18 | 6.20 | sunny | Train |
| 2012-05-11 | 6.00 | sunny | Train |
| 2012-02-12 | 5.80 | sunny | Eval |
| 2012-02-19 | 6.20 | overcast | Eval |
| 2012-03-31 | 6.00 | overcast | Eval |
| 2012-05-26 | 6.30 | sunny | Eval |

Table 3: Ablation of using text at inference.

| Method | QEOxford | Oxford | NCLT |
|---|---|---|---|
| vanilla | 0.83/1.12 | 2.67/1.25 | 1.46/2.80 |
| w pred text | 0.75/1.02 | 2.60/1.21 | 1.41/2.70 |
| w gt text | 0.74/0.93 | 2.58/1.07 | 1.12/2.40 |
| **ours** | 0.75/1.03 | 2.59/1.19 | 1.40/2.62 |

Table 4: Classification accuracy of position/orientation on different datasets.

| Dataset | Accuracy |
|---|---|
| QEOxford | 99.19%/97.64% |
| Oxford | 98.99%/97.52% |
| NCLT | 98.44%/85.50% |

**More visualization.** To further dissect the performance on the Oxford and NCLT datasets, Fig. 1 and Fig. 2 provide trajectory visualizations and cumulative error distribution curves. The results of sequences 17-13-26-39 (Oxford) and 2012-02-19 (NCLT) are provided for comparison, respectively. GTR-Loc's estimated trajectory adheres closely to the ground truth throughout different sequences. Even within structurally complex or repetitive areas (as marked by blue boxes), GTR-Loc maintains consistent localization. Methods like SGLoc or LiSA exhibit jumps in these areas. The cumulative error distribution curves also demonstrate GTR-Loc's leading performance across most error ranges. Our curves for both position and orientation errors lie predominantly above those of other methods, suggesting overall lower error magnitudes.

**Localization Failure Cases.** The model's failures are confined to rare, extreme scenarios where LiDAR data contain almost no distinctive geometric features. For example, on a very long street in

the Oxford dataset, the building facades, streetlights, and other structures are completely identical. In these few instances (<1% of the trajectory), the teacher model succeeds by leveraging geospatial text, an external cue that the student lacks. While these specific failures can produce large errors, their statistical impact becomes negligible when averaged over the entire dataset. In the vast majority of cases, the student effectively utilizes subtle geometric cues, achieving performance nearly identical to the teacher's.

## B    Limitations and Future Work

**Limitation.** Despite its promising results, GTR-Loc has limitations that highlight areas for future research. While our distillation approach effectively eliminates the need for text processing during inference, a limitation is that leveraging text directly at inference time demonstrably achieves better performance. Our experiments demonstrate that using ground-truth text, rather than erroneously predicted text, is more beneficial for accurate localization.

**Future Work.** Consequently, our future work will concentrate on exploring methods for generating more accurate textual descriptions without using ground truth poses during the inference phase. The aim is to significantly enhance LiDAR-text localization precision by effectively harnessing these improved textual cues at the point of decision-making.

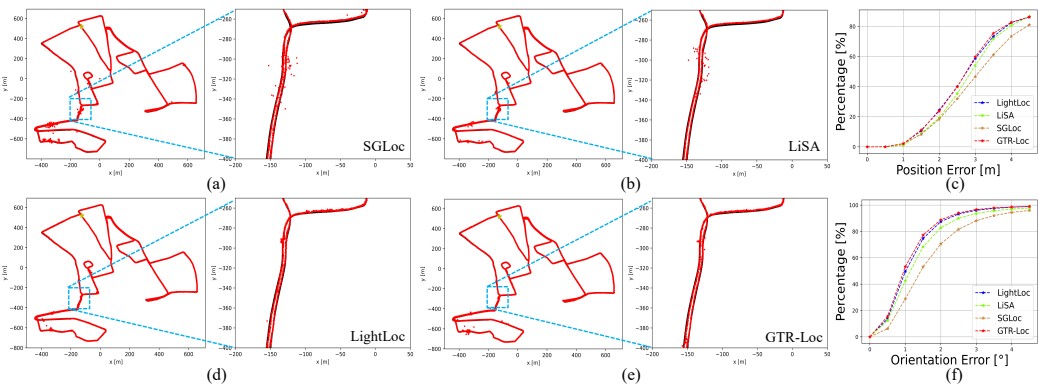

Figure 1: Visual comparisons on Oxford. (a) (b) (d) (e): predicted trajectories (red) overlaid on ground truth (black); a star marks the starting position, and the blue box highlights a challenging road segment. (c) (f): cumulative error distribution curves for position (top) and orientation (bottom).

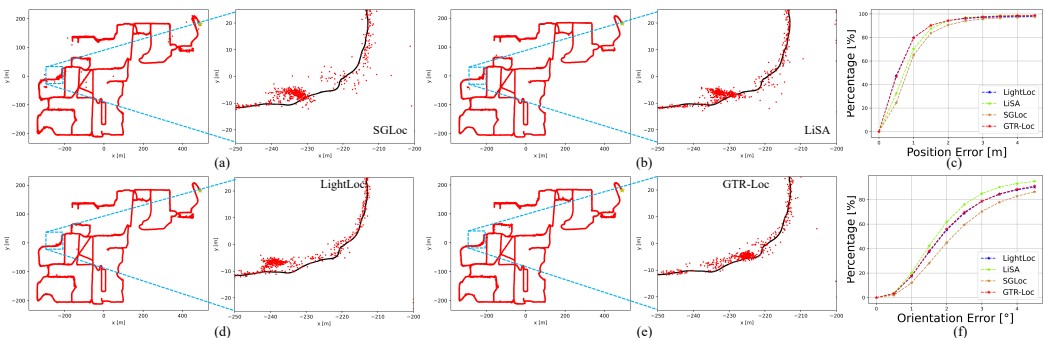

Figure 2: Visual comparisons on NCLT. The notations follow Fig. 1.

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
