# OpenReview forum: "GTR-Loc: Geospatial Text Regularization Assisted Outdoor LiDAR Localization"
_NeurIPS.cc/2025/Conference — NeurIPS 2025 poster_

### Official Review · Reviewer_YNwr · 2025-06-09

**Clarity:** 2
**Significance:** 3
**Originality:** 3
**Rating:** 4
**Confidence:** 3

**Summary:**

This paper introduces GTR-Loc, a method for outdoor LiDAR localization designed to address the problem of geometric ambiguity in scene coordinate regression methods. The core idea is to regularize the learning process using structured, pose-aware textual descriptions. The authors propose a "Geospatial text generator" that creates formatted text (eg. "District 1, West") from ground truth pose information during training. This text is then refined by a LiDAR-Anchored Text embedding refinement module, which conditions the text embedding on the current LiDAR scan to create a view-specific representation. To make the method practical for inference where ground truth pose is unavailable, the paper proposes a distillation strategy. This involves training a LiDAR-only "student" model to mimic the predictions of the text-assisted "teacher" model. Experiments on standard datasets like QEOxford, Oxford, and NCLT show that GTR-Loc outperforms existing state-of-the-art methods.

**Questions:**

- The core of your method is using discretized pose information as a regularizer. Could you elaborate on why this is framed as "text-assisted" localization, which implies a connection to natural language understanding, rather than as learning with auxiliary supervision from discretized pose labels? My score could be raised if you can provide a strong justification for why the "language" aspect is fundamental, beyond simply being a carrier for the discrete IDs. Alternatively, re-framing the contribution around a novel auxiliary task and distillation scheme would also clarify the paper's core novelty.

- Could you justify the use of a pre-trained CLIP text encoder for your simple, template-based strings? Have you considered or run experiments with a simpler baseline where the district and direction IDs are encoded via a simple, randomly initialized embedding layer and then concatenated or fused? Demonstrating that the CLIP encoder provides a significant, non-trivial benefit over a much simpler embedding scheme would strongly justify your design choice. If a simpler method performs comparably, it would be a valuable finding to include, potentially simplifying the model.

- Table 7 shows a small but consistent performance gap between the full "teacher" model (using ground truth text at inference) and the distilled "student" model. Could you provide more insight into this gap? For example, are there specific scenarios, environments (eg. open areas vs. dense city blocks), or types of ambiguity where the distilled model is less effective than the teacher?

**Ethical Concerns:**

["NO or VERY MINOR ethics concerns only"]

**Final Justification:**

Overall, the paper proposes an interesting idea centered around using embeddings of places to resolve ambiguity in SCR-based localization. My concerns regarding the performance of CLIP vs randomly initialized embeddings was satisfactorily addressed. I still feel that the "semantic" nature of the text is not really apparent when formulated as [district ID, direction ID] and could be better utilised. My final rating is Borderline Accept.

**Limitations:**

Yes

**Quality:**

3

**Strengths And Weaknesses:**

**Strengths**
- The paper tackles the well-known and significant problem of localization ambiguity in large-scale environments, where geometrically similar but spatially distinct places can confuse SCR-based methods. The motivation is clear and the proposed direction of using additional context is sound.

- The MRD strategy is interesting. It provides an elegant solution to the problem of needing to generate text at inference time. This teacher-student framework makes the approach practical for real-world deployment and is a good contribution in this context.

- The method demonstrates state-of-the-art results on several challenging benchmarks. The reported 9.64%/8.04% improvement in position/orientation accuracy over a strong baseline (LightLoc) on the QEOxford dataset is good and highlights the effectiveness of the proposed regularization.

- The authors have conducted a comprehensive set of ablation studies that validate the contribution of each component of GTR-Loc. The experiments effectively demonstrate the benefits of using the proposed geospatial text over scene descriptions (Table 4), the effectiveness of the LATER module's Transformer-based fusion (Table 6), and the impact of the MRD strategy (Table 7).

**Weaknesses**
- The main weakness lies in the framing of the core contribution as "geospatial text regularization." The "text" generated by GTG is not natural language but rather a simple, structured concatenation of two discrete labels (district ID and direction ID). While effective, this feels more like a very clever form of feature engineering or an auxiliary task with discretized pose labels than a true integration of language and vision. Invoking the representation power of vision-language models like CLIP seems somewhat overstated when the input vocabulary is so limited and structured.

- Following the point above, the paper does not sufficiently justify the use of a powerful, pre-trained text encoder from CLIP for such simple, template-based strings. The semantic richness that a CLIP encoder is designed to capture seems like overkill for inputs like "District 99, West-Southwest." It is unclear if a much simpler approach, such as using separate learnable embedding layers for the district and direction IDs, would have performed similarly.

- While the paper includes an ablation on the number of districts and directions (Table 5), the discussion is brief. The choice of z=100 and d=16 is presented as optimal, but there is little analysis of the trade-offs. For instance, how does performance degrade in very large-scale maps where 100 districts might be too coarse? Conversely, how sensitive is the model to overfitting if the granularity is too fine?

---

> ### Author Rebuttal · Authors · 2025-07-30
>
> Thank you for your insightful comments.
> We have addressed each point and will integrate all corresponding revisions into the final version of the paper.
>
> **Response to W1\&Q1.**
> (1) *Why "text-assisted" localization?*
> The focus of our method is not on processing natural language, but on learning a novel geospatial text regularization to solve localization ambiguities.
> We use the term "text" because we convert discrete geospatial labels into text strings (e.g., "District 50, North"), which are then processed within the text modality.
> This textification is critical because it leverages the rich semantic knowledge embedded in the CLIP text encoder.
> If we use discrete IDs (e.g., 50) as categorical labels instead, the model needs to learn complex relationships between these meaningless symbols and point features from scratch.
> This not only presents significant optimization challenges but also leads to overfitting.
> The model cannot comprehend inherent spatial relationships like the adjacency of "District 50" and "District 51" or the opposition of "North" and "South."
>
> In contrast, by converting the labels into text with semantic meaning (e.g., "District 50, North"), we leverage the vast world knowledge and text understanding capabilities embedded in pre-trained VLMs.
> The powerful text encoders of traditional VLMs (e.g., CLIP) have excellent robustness and generalization capabilities for text.
> CLIP understands that "North" and "Northeast" are semantically related.
> This significantly reduces the dependency on task-specific labeled data, and because of the flexibility of text, a fixed network architecture can easily adapt to different datasets or changing data categories.
> Hence, the proposed geospatial text is not a simple feature engineering.
>
> (2) *Re-framing the contribution.*
> We acknowledge that the framing could be clearer.
> In the final version, we will refine our paper to better articulate that we use CLIP to process our novel geospatial text, rather than proposing a new natural language processing.
>
> **Response to W2\&Q2.**
> (1) *Why use CLIP?*
> We use a pre-trained CLIP encoder to leverage its rich prior, even for simple, structured text.
> A randomly initialized embedding layer will treat district and direction IDs as independent and arbitrary tokens.
> Hence, it needs to learn relationships from scratch, leading to less efficiency and poor generalization.
> In contrast, CLIP already understands that "North" and "South" are opposites and "Northeast" is an intermediate geographical concept.
> This meaningful relational structure can adapt to scene changes (e.g., adding districts or changing datasets) without retraining.
> CLIP can also better handle the composition of tokens like "District 50" and "North," rather than simple concatenation.
>
> (2) *Replace CLIP with embedding layers.*
> We conduct new ablation studies by replacing the CLIP text encoder with embedding layers (ELs).
> Specifically, we learn two separate, learnable embedding layers (nn.Embedding) initialized from scratch: one for the 100 district IDs and another for the 16 direction IDs.
> The resulting vectors are concatenated and passed through an MLP to match the feature dimension of the original CLIP embedding.
> The results shown below indicate that using simple learnable embeddings only leads to a small performance improvement.
> Although it is still effective, its performance is noticeably inferior to the full GTR-Loc that uses CLIP.
> Hence, using a powerful, pre-trained CLIP text encoder is not overkill.
>
> |Method|QEOxford|Oxford|NCLT|
> |-|-|-|-|
> |vanilla|0.83/1.12|2.67/1.25|1.46/2.80|
> |w ELs|0.80/1.10|2.64/1.23|1.44/2.75|
> |w CLIP (ours)|0.75/1.03|2.59/1.19|1.40/2.62|
>
> **Response to W3.**
> (1) *Coarse partitioning on large-scale maps.*
> Coarser partitions create overly large districts that can encompass multiple visually similar yet geographically separate locations.
> Then, the text would lose its power to disambiguate.
> Since the dataset scale cannot be increased, we use fewer partitions to illustrate the same situation.
> The original results in Table 5 show that as partitions decreases from z=100 to z=49, localization accuracy decreases.
> As shown below, we conduct new ablation studies with fewer partitions (z=25 and z=9).
> The results indicate that coarser partitions can not provide effective discriminative cues.
> However, as the district number is a hyperparameter, it can be scaled with the map size.
>
> (2) *Sensitivity to fine-grained partitioning.*
> Excessively fine partitions risk overfitting due to very sparse data in each district, which can turn the textual cue into a rigid lookup key instead of a soft regularizer. However, our design demonstrates robustness to this.
> Results in Table 5 and the table below, respectively, show that as we increase partitions to z=144, and even to extreme values like z=2500 and z=10000, performance plateaus rather than degrades sharply.
> It suggests that the model is not overly sensitive to overfitting.
> This resilience stems from the regularizing effect of MRD distillation, which effectively smooths out noise from the sparse data within these fine-grained partitions.
>
> |Method|QEOxford|Oxford|NCLT|
> |-|-|-|-|
> |z=9,d=16|0.82/1.11|2.64/1.24|1.44/2.74|
> |z=25,d=16|0.81/1.11|2.64/1.25|1.43/2.73|
> |z=100,d=16|0.75/1.03|2.59/1.19|1.40/2.62|
> |z=2500,d=16|0.82/1.11|2.64/1.24|1.43/2.76|
> |z=10000,d=16|0.82/1.15|2.63/1.28|1.45/2.87|
>
> **Response to Q3.**
> (1) *Performance gap analysis.*
> The small performance gap shows that knowledge distillation successfully transfers the teacher's reasoning, not just its results.
> By mimicking the teacher's text-regularized approach, the student learns to resolve subtle geometric ambiguities.
> This is highly effective on datasets focused on geometric ambiguities like QEOxford/Oxford, where the student's position error gap is minimal (as low as 1.33%/0.39%).
> The gap is larger on NCLT, likely due to less stable data from its Segway collection platform.
> Across all datasets, the consistent gap in orientation error (more than 5%) suggests that distilling knowledge for precise orientation is more challenging than for position.
>
> (2) *Failure scenario for the student.*
> The student model's failures are confined to rare, extreme scenarios where LiDAR data contain almost no distinctive geometric features.
> For example, on a very long street in the Oxford dataset, the building facades, streetlights, and other structures are completely identical.
> In these few instances (<1% of the trajectory), the teacher model succeeds by leveraging geospatial text, an external cue that the student lacks.
> While these specific failures can produce large errors, their statistical impact becomes negligible when averaged over the entire dataset.
> In the vast majority of cases, the student effectively utilizes subtle geometric cues, achieving performance nearly identical to the teacher's.

---

> > ### Comment · Reviewer_YNwr · 2025-08-07
> >
> > Thank you for the detailed response. The experiments with the embedding layer compared to Clip are very informative It seems, as I expected, the ELs do quite well but CLIP shows a slight advantage. Another interesting experiment (not necessary to include now) would be to look at the embedding space of CLIP and see whether it's possible to extract the rough spatial relations directly from the [district ID, direction ID]'s. This would give proof that CLIP provides meaningful spatial information from the text.
> >
> > The partitioning experiments are also interesting as they show the robustness to the partitioning. It would be good to include both tables in the paper.

---

> > > ### Author Response · Authors · 2025-08-08
> > > **Response to Reviewer Feedback**
> > >
> > > Thank you for the insightful feedback and excellent suggestions for the paper.
> > >
> > > We're glad you found the comparison insightful. It affirms our central finding: CLIP's pre-trained knowledge provides a consistent edge in performance and zero-shot generalization, which is precisely why we chose it.
> > >
> > > A t-SNE visualization of spatial relationships, though not possible to include here due to rebuttal limitations, will be added to the revised paper. We provide a case study on QEOxford to validate our CLIP embedding space. We correlate embedding distance (cosine) with geographic distance (m) and find the expected positive correlation. For example, the adjacent pair (43 vs. 44) has an embedding distance of just 0.18, proving CLIP's representation is more meaningful and geographically coherent than a standard embedding.
> > >
> > > |District|Geographic Distance|CLIP Embedding Distance|ELs Embedding Distance|
> > > |-|-|-|-|
> > > |43 vs. 44|100|0.18|0.45|
> > > |43 vs. 53|150|0.21|0.62|
> > > |43 vs. 73|450|0.52|0.55|
> > > |43 vs. 93|750|0.71|0.73|
> > >
> > > We also agree that the partitioning experiments are important for demonstrating the model's robustness. Based on the reviewer's valuable feedback, we will be sure to include both tables in the final version of the paper.

---

### Official Review · Reviewer_zga8 · 2025-06-20

**Clarity:** 3
**Significance:** 2
**Originality:** 3
**Rating:** 5
**Confidence:** 4

**Summary:**

This paper introduces GTR-Loc, a text-assisted LiDAR localization framework that addresses the critical problem of localization ambiguities in outdoor environments. The method enhances Scene Coordinate Regression (SCR) by integrating geospatial text descriptions to distinguish between geometrically similar but spatially distinct locations. The framework consists of three main components: a Geospatial Text Generator (GTG) that creates discrete pose-aware text descriptions, a LiDAR-Anchored Text Embedding Refinement (LATER) module that produces view-specific embeddings, and a Modality Reduction Distillation (MRD) strategy that enables LiDAR-only inference.

**Questions:**

1. The discrete geospatial text, while robust, may be overly simplistic. The fixed district/direction partitioning might not capture all relevant localization cues, particularly in environments with rich semantic content that could aid disambiguation.

2. One of the important factor in LiDAR Localization is efficiency. The reviewer did not find any comparison to previous methods in main paper and Appendix. It seems like there are more complicated modules involving self-/cross-attention that might make the method less efficient. It will be better to provide some quantitative results on this (for example, processing fps during evaluation).

3. The benefit of text comes from the distillation loss. In Table 7, the results show that with distillation, the performance is similar to the performance using text in testing. This leads to two questions: a) using text is actually using an approximate ground truth location. But seems like the performance improvement against previous methods is not that significant. Does it indicate that progress can be made in the way to integrate text? b) The reviewer is wondering how the performance is to simply cut the MRD module and train the network for longer time because seems like the information provided in text is also provided in ground truth localization labels used for training. The LATER module makes it explicit and utilize text embedding to help training. This **might** only accelerate training process. Thus the reviewer hope to see more results to support the claims.

**Ethical Concerns:**

["NO or VERY MINOR ethics concerns only"]

**Final Justification:**

During rebuttal, the authors well resolved the reviewer's concerns about (1) the text descriptions. (2) computational cost. (3) convergence. (4) understanding of the role that text descriptions play in the pipeline.

The reviewer raise the score to 5.

**Limitations:**

Please see Questions part and Weakness part.

**Paper Formatting Concerns:**

N.A.

**Quality:**

2

**Strengths And Weaknesses:**

## Strengths
1. Novel Problem Formulation and Solution
The paper identifies a fundamental limitation in current LiDAR localization methods - the inability to distinguish between geometrically similar locations. The proposed text-assisted approach is well-motivated.
2. Extensive experiments.
3. The authors provide codes in the supplementary materials.

## Weakness
1. Limited Scope of Text Information.
The discrete geospatial text, while robust, may be overly simplistic. The fixed district/direction partitioning might not capture all relevant localization cues, particularly in environments with rich semantic content that could aid disambiguation.
2. Lack of Efficiency Comparison. One of the important factor in LiDAR Localization is efficiency. The reviewer did not find any comparison to previous methods in main paper and Appendix. It seems like there are more complicated modules involving self-/cross-attention that might make the method less efficient. It will be better to provide some quantitative results on this (for example, processing fps during evaluation).
3. The benefit of text comes from the distillation loss. In Table 7, the results show that with distillation, the performance is similar to the performance using text in testing. This leads to two questions: a) using text is actually using an approximate ground truth location. But seems like the performance improvement against previous methods is not that significant. Does it indicate that progress can be made in the way to integrate text? b) The reviewer is wondering how the performance is to simply cut the MRD module and train the network for longer time because seems like the information provided in text is also provided in ground truth localization labels used for training. The LATER module makes it explicit and utilize text embedding to help training. This **might** only accelerate training process. Thus the reviewer hope to see more results to support the claims.

---

> ### Author Rebuttal · Authors · 2025-07-30
>
> Thank you for your insightful comments.
> We have addressed each point and will integrate all corresponding revisions into the final version of the paper.
>
> **Response to W1\&Q1.**
> (1) *Inconsistent and indirect descriptive texts.*
> Human-like text generation, such as TOD3Cap in Figure 3, tends to describe target-environment-relationship, such as vehicles or vegetation.
> These free-form scene descriptions are inherently unstable; dynamic objects and environmental changes lead to inconsistent characterizations of the same location over time, providing an unreliable description.
> Furthermore, these descriptions offer only indirect cues for 6-DoF pose estimation, with irrelevant details acting as noise that can impede learning.
> This challenge is compounded by the sparse, geometric-only nature of LiDAR point clouds, which, unlike images, lack rich texture and color.
>
> (2) *Stable and direct GTG.*
> In contrast, our GTG produces text (e.g., "District 99, West-Southwest") that yields a standardized representation of universal pose information, creating a signal that is stable, repeatable, and directly relevant to pose estimation.
> The text's invariance to dynamic content provides the consistency needed to resolve localization ambiguities via regularization.
> Then, the LATER module refines GTG-generated text by fusing it with the current point cloud's geometry, creating viewpoint-specific embeddings that enrich the final training feature with diverse visual-geometric information beyond the core pose.
>
> (3) *Experimental validation and proof.*
> In Table 4, we demonstrate that GTG is more effective than free-form scene descriptions.
> We also present this table here.
> We replace GTG with a SOTA 3D scene captioning model, TOD3Cap, to generate rich text.
> This free-form text mainly describes object motion, appearance, relationship, and environment.
> However, using free-form descriptions from TOD3Cap yields no significant performance gain over the vanilla model.
> Our template-based GTG leads to significant improvements.
>
> |Method|QEOxford|Oxford|NCLT|
> |-|-|-|-|
> |vanilla|0.83/1.12|2.67/1.25|1.46/2.80|
> |TOD3Cap|0.84/1.15|2.66/1.24|1.47/2.78|
> |GTR-Loc|0.75/1.03|2.59/1.19|1.40/2.62|
>
> **Response to W2\&Q2 Time evaluation.**
> Our method is real-time and highly efficient during inference.
> On the Oxford/QEOxford datasets, the average processing time per sample is 29ms (34 FPS), and on the NCLT dataset, it is 48ms (21 FPS).
> These speeds are well within the respective scanning frequencies of 20 Hz for Oxford/QEOxford and 10 Hz for NCLT, highlighting the model’s ability to maintain high accuracy during real-time operation.
> We also report the runtime comparison with SCR baselines, as shown in the table below.
> Our runtime outperforms SGLoc and LiSA, while being on par with LightLoc.
> However, our superior performance demonstrates the advantage of GTR-Loc in both accuracy and efficiency.
>
> Our model's complex self/cross-attention modules are confined to the teacher network (blue flow, Fig. 2), which is used only during offline training.
> This design ensures that our final inference model, obtained via MRD distillation, is a lightweight, LiDAR-only network (red flow, Fig. 2) with efficiency comparable to baselines like LightLoc.
> To better highlight this, we will move the detailed time analysis from the appendix into the main text.
>
> |Method|QEOxford|Oxford|NCLT|
> |-|-|-|-|
> |SGLoc|38ms|38ms|75ms|
> |LiSA|38ms|38ms|75ms|
> |LightLoc|29ms|29ms|48ms|
> |GTR-Loc|29ms|29ms|48ms|
>
> **Response to W3\&Q3.**
> (1) *Is text integration the bottleneck?*
> First, considering our text input as an "approximate ground truth location" may be a misunderstanding.
> Taking the QEOxford dataset as an example, its area is approximately 1000m x 1500m, which we divide into a 10x10 grid.
> This means each geospatial district is about 100m x 150m in size.
> For a localization task requiring meter- or even sub-meter-level accuracy, a regional label spanning hundreds of meters is far from an "approximate ground truth."
> In addition, our framework is SCR, which predicts world coordinates for each input point.
> The ground truth label is coordinate labels rather than pose labels.
>
> Second, we would like to respectfully clarify that GTR-Loc achieves superior performance on all three datasets.
> Our method outperforms all baselines on QEOxford, achieves SOTA position accuracy on Oxford and NCLT, and ranks second in orientation estimation.
> Specifically, on QEOxford, GTR-Loc (0.75m/1.03°) achieves a significant 9.64%/8.04% improvement in both position and orientation over the previous SOTA, LightLoc (0.83m/1.12°).
> GTR-Loc consistently outperforms LightLoc on both the Oxford (2.59m/1.19° vs. 2.67m/1.25°) and NCLT (1.40m/2.62° vs. 1.46m/2.80°) datasets.
> The absolute improvement in position accuracy over previous leading SCR methods, LiSA and LightLoc, is significant.
> The results in Table 7 mainly demonstrate the effectiveness of our MRD distillation, with the LiDAR-only model achieving performance nearly on par with the LiDAR-text version.
>
> (2) *Does text regularization merely accelerate training?*
> Our method does not simply accelerate training; it fundamentally alters the learning process.
> Traditional SCR methods, using per-point loss, can get trapped in local minima when encountering geometrically similar scenes.
> To address this, our method introduces text regularization as a unified, global constraint.
> For instance, the text "District 50, North" mandates that all predicted coordinates are macroscopically consistent with this region, creating a more favorable optimization landscape when combined with the original per-point loss.
> Therefore, this text guidance is not merely a training accelerator but a fundamental tool for resolving localization ambiguities.
>
> |Method|QEOxford|Oxford|NCLT|
> |-|-|-|-|
> |w/o MRD (normal epoch)|0.83/1.12|2.67/1.25|1.46/2.80|
> |w/o MRD (3x epoch)|0.84/1.11|2.66/1.24|1.46/2.81|
> |w MRD (normal epoch)|0.75/1.03|2.59/1.19|1.40/2.62|
> |w MRD (3x epoch)|0.75/1.02|2.60/1.18|1.40/2.61|
>
> We conduct new ablation studies by taking the baseline model (the LiDAR-only network architecture without a teacher and MRD) and extending its training time by 3x (from 25 to 75 epochs on QEOxford/Oxford and 30 to 90 epochs on NCLT).
> As shown in the table, simply training for longer yields nearly no performance gain without MRD.
> It comes nowhere close to matching the results of GTR-Loc w MRD (normal epoch) under its standard training schedule.
> In addition, training GTR-Loc for a longer time also will not improve performance because the network has already converged.
> This provides strong evidence that our method reshapes the learning process, rather than merely accelerating it.

---

> > ### Comment · Reviewer_zga8 · 2025-08-01
> > **Please clarify some remaining concerns on Q2 and Q3.**
> >
> > Thanks for the detailed rebuttal and the reply to the questions. The reviewer has some remaining concerns regarding Q2 and Q3.
> >
> > For Q2, could the authors provide statistic about training time comparison?
> >
> > For Q3, the authors claim that "This means each geospatial district is about 100m x 150m in size.". Could we consider the text information as a good initialization point to narrow down the search space?

---

> > > ### Author Response · Authors · 2025-08-02
> > > **Response to remaining concerns on Q2 and Q3.**
> > >
> > > Thank you for your time and for providing further feedback on our rebuttal.
> > > We have carefully considered your remaining concerns and hope our responses can fully address them.
> > >
> > > **Response to Q2:**
> > > The entire training process for our full GTR-Loc framework, which includes all components in Figure 2, takes approximately 4 hours on  QEOxford/Oxford and 3 hours on NCLT.
> > > The training time comparison for SCR baselines is shown below.
> > > All experiments are conducted on a single NVIDIA RTX 4090 GPU.
> > > Our training time is significantly shorter than that of SGLoc and LiSA.
> > > Compared to the existing SOTA LightLoc, our training time is a bit longer because we include a LiDAR-text SCR, a LiDAR-only SCR, and a distillation process.
> > > The critical finding, however, is not the training speed but the final performance.
> > > Our performance is significantly superior, while our inference efficiency is on par with LightLoc.
> > >
> > > |Method|QEOxford|Oxford|NCLT|
> > > |-|-|-|-|
> > > |SGLoc|50h|50h|42h|
> > > |LiSA|53h|53h|44h|
> > > |LightLoc|1h|1h|1h|
> > > |GTR-Loc|4h|4h|3h|
> > >
> > > **Response to Q3:**
> > > Yes, that's an excellent and insightful way to view it.
> > > Our method is based on LiDAR SCR, a framework that is powerful but known to suffer from localization ambiguity due to similar geometric structures in different locations.
> > > Lacking sufficient guiding information, a standard SCR is prone to converging to a local optimum, irrespective of the training time.
> > > To solve this fundamental problem of ambiguity, we introduce the geospatial text.
> > > It acts as a powerful regularizer that guides the optimization process away from incorrect solutions.
> > > As the reviewer insightfully suggests, this text can be considered as a powerful prior to "narrow down the search space".
> > > For example, "District 99" provides a strong initial hypothesis that constrains localization and prevents large-scale errors.

---

> > > > ### Comment · Reviewer_zga8 · 2025-08-03
> > > >
> > > > Thank you for making Q2 clearer. For Q3, as the text information can be regarded as a good initialization point to narrow down the search space, one follow-up question would be why we do not need it during testing. Could you further clarify this point?

---

> ### Author Response · Authors · 2025-08-04
> **Response to remaining concerns on Q3.**
>
> **Response to Q3:** Thank you for your further feedback on our rebuttal. We appreciate the opportunity to clarify this point and will make it clearer in the revision.
>
> Using geospatial text for inference may be impractical because generating it requires gt poses or pose-relevant information, the very thing we aim to estimate. To solve this, we design the MRD distillation. We use a teacher-student framework for training. The teacher (LiDAR+Text) learns to resolve ambiguities, while the student (LiDAR-only) mimics the teacher's outputs. Through distillation, the LiDAR-only student inherits the teacher's text-assisted knowledge, learning to "narrow the search space" on its own using only point clouds.
>
> Importantly, our deployed model (a distilled, LiDAR-only GTR-Loc) requires no text during inference and uses only a single LiDAR scan. However, we further investigate the case where text is used during inference. We first test a hypothetical scenario where gt poses are available to generate perfect text for the teacher model (blue flow, Fig. 2) at inference. The ablation study in Table 7 demonstrates that the distilled model's performance (row 4 of the table below) is nearly identical to the teacher (row 2 of the table below) that uses perfect text during testing. This proves that our MRD successfully transfers the vast majority of the text's benefits into the LiDAR-only model.
>
> ||Method|QEOxford|Oxford|NCLT|
> |-|-|-|-|-|
> |1|vanilla|0.83/1.12|2.67/1.25|1.46/2.80|
> |2|Teacher w text (gt)|0.74/0.93|2.58/1.07|1.12/2.40|
> |3|Teacher w text (pred)|0.75/1.02|2.60/1.21|1.41/2.70|
> |4|GTR-Loc|0.75/1.03|2.59/1.19|1.40/2.62|
>
> Since gt poses are usually unavailable during inference, we then try to train an auxiliary classification network to predict district and direction categories at test time. The predicted categories are used as district and direction IDs for text generation. The network comprises a LightLoc encoder and 4 FC layers. The results shown in the appendix (row 3 of the table above) indicate that the teacher with predicted text inputs is not effective. It yields virtually no performance benefits at the cost of higher computational complexity and longer test times. Inaccurate position and orientation classification (the table below) leads to wrong text generation, thus resulting in poor localization.
>
> |Dataset|Pos./Ori. Acc|
> |-|-|
> |QEOxford|99.19\%/97.64\%|
> |Oxford|98.99\%/97.52\%|
> |NCLT|98.44\%/85.50\%|

---

> > ### Comment · Reviewer_zga8 · 2025-08-05
> >
> > Thanks for the clarification. All the concerns are resolved and the reviewer is willing to raise the score.

---

> > > ### Author Response · Authors · 2025-08-06
> > >
> > > Thank you so much for the update! We are delighted to hear that our clarifications have resolved the reviewer's concerns and earned a higher score. We sincerely thank the reviewer for their valuable comments and final approval. All responses will be updated in the revised manuscript.

---

### Official Review · Reviewer_ispn · 2025-07-02

**Clarity:** 3
**Significance:** 3
**Originality:** 3
**Rating:** 4
**Confidence:** 4

**Summary:**

This paper introduces GTR-Loc, a text-assisted LiDAR localization framework that enhances scene coordinate regression (SCR) by integrating geospatial text regularization. The key contribution is the novel use of discrete pose-aware textual descriptions to mitigate localization ambiguities in large-scale outdoor environments. This design augments geometric data with structured text, providing additional spatial cues. To handle intra-region variability, the authors propose a LiDAR-Anchored Text Embedding Refinement (LATER) module that adapts text embeddings based on local point cloud features. The model is trained using a dual-regression architecture with a Modality Reduction Distillation (MRD) strategy to enable efficient LiDAR-only inference. The authors conduct experiments showing that the proposed method consistently improves localization robustness. Furthermore, the paper presents extensive ablation analyses, demonstrating that the proposed components contribute meaningfully to resolving ambiguity in visually similar scenes.

**Questions:**

The paper is well-executed, and the results are valuable. My questions focus on clarifying some details in experiments.

Vision-language fusion. The LATER module uses a Transformer to refine text embeddings with point-cloud features, yet its conceptual relation to existing prompt-tuning or cross-attention paradigms (e.g., CoOp, BLIP, Flamingo) is not discussed. Please clarify (i) which design choices are novel versus inherited, and (ii) why LATER is preferable to directly adopting those prior fusion schemes.

Result variability. The experimental results are promising, but all reported numbers appear to be single-run averages. Given that some gains (e.g., over LightLoc) are modest, can the authors report variance estimates or confidence intervals across multiple random seeds or splits, especially for QEOxford and NCLT?

Adaptivity of discretization. The use of fixed spatial partitioning (z = 100 districts, d = 16 directions) is central to GTG, yet Table 5 suggests diminishing returns beyond certain granularity. Have the authors considered data-driven or learnable discretization schemes? Additionally, are there qualitative failure cases that arise from coarse or misaligned partitions?

**Ethical Concerns:**

["NO or VERY MINOR ethics concerns only"]

**Limitations:**

Yes

**Paper Formatting Concerns:**

No formatting issues were identified.

**Quality:**

3

**Strengths And Weaknesses:**

### Strengths
Structured text regularization. The method leverages discrete, pose-aware textual descriptions as a form of geospatial regularization to complement geometric features in scene coordinate regression. This formulation offers an alternative pathway to reduce localization ambiguities, and represents a reasonable extension of multimodal integration in the context of LiDAR-based localization.

Modular and generalizable design. The proposed framework is composed of well-defined modules—namely, the Geospatial Text Generator, LATER refinement module, and the Modality Reduction Distillation strategy—which collectively form a coherent and extensible pipeline. This modularity not only aids interpretability but also facilitates adaptation to related multimodal perception tasks beyond localization.

Strong empirical performance. The paper presents comprehensive experimental results showing consistent improvements over a range of existing LiDAR localization baselines across multiple benchmarks. Compared to prior work on the same task, the proposed method demonstrates stronger overall performance, supported by clear quantitative evidence. This level of empirical validation reinforces the practical effectiveness of the approach and lends credibility to its contributions.

### Weaknesses

Assumptions on Spatial Discretization. The proposed framework relies on predefined offline discretization of spatial regions and orientation bins, which, while effective under controlled settings, may pose challenges in dynamic or evolving environments. The paper does not explicitly address how the method generalizes under long-term environmental changes or large-scale deployment scenarios, potentially limiting its applicability in real-world conditions where map partitioning is not fixed or stable.

Limited Comparison in Multimodal Design. While the LATER module presents a promising integration of point cloud features and geospatial text, its relation to prior multimodal fusion paradigms—such as prompt-based vision-language models (e.g., CoOp, BLIP)—is not thoroughly explored. This lack of comparative positioning slightly weakens the clarity around the novelty boundary of the method and may leave readers uncertain about its conceptual advances relative to existing multimodal architectures.

---

> ### Author Rebuttal · Authors · 2025-07-30
>
> Thank you for your insightful comments.
> We have addressed each point and will integrate all corresponding revisions into the final version of the paper.
>
> **Response to W1 Assumptions on Spatial Discretization.**
> (1) *Long-term environmental changes.*
> Although our method relies on a predefined map partition, our partitioning scheme is designed to be robust to environmental changes.
> The partition is not based on semantic features of the environment (like roads or buildings), but on an abstract, universal coordinate grid.
> Hence, with long-term environmental changes, such as construction and demolition, seasonal and vegetation changes, the partition is still stable.
> Results on the NCLT dataset, known for its data collection spanning several months, also highlight our method's robustness to long-term variations.
>
> (2) *Large-scale deployment.*
> Our design is also methodologically scalable for large-scale deployment.
> While we use a 10x10 uniform grid in this paper, it is not a limitation of the method.
> For larger deployments (e.g., an entire city), the strategy can be seamlessly extended.
> For example, we can adopt standard practices from geospatial applications, such as a hierarchical UTM grid, to programmatically partition an area of any size.
> In addition, our ablation study (Table 5) also demonstrates the robustness of our approach (e.g., increasing from 100 to 144 districts).
> This is critical for scalability, as a reasonable, predefined partition is sufficient, making the framework practical for real-world use.
>
> **Response to W2\&Q1 Vision-language fusion.**
> (1) *Novelty of LATER.*
> The novelty of our approach stems from both its design purpose and specific architectural details.
> Previous VLMs bridge vision and language via methods like CoOp's learnable prompts, BLIP's contrastive-fusion learning, or Flamingo's gated cross-attention for injecting visual tokens into frozen LLMs.
> They are designed for image understanding tasks such as image captioning or zero-shot classification, while our task is LiDAR localization.
> LATER is specifically designed to dynamically refine a pose-aware text embedding based on the unique geometric features of the current LiDAR scan, thus regularizing localization.
>
> In addition, our novel module details are also different from previous VLMs.
> Following the well-established paradigm set by VLMs, LATER also utilizes a Transformer architecture and learnable prompts.
> However, LATER does not simply fuse the point cloud and text; instead, it independently trains a Transformer to dynamically refine learnable prompts ${v_n}$ using point features $F_p$.
> Subsequently, it concatenates these visually-tuned prompts $v'_n$ with a static text embedding that contains coarse pose information from our custom generator.
>
> (2) *Advantages of LATER.*
> Our LATER is preferable because it is a tailored solution to solve view-dependent ambiguities in LiDAR localization.
> Directly adopting a fusion module from models like CoOP or Flamingo would be suboptimal for our task.
> They are designed to fuse dense images with rich, descriptive texts, aiming to maximize the semantic understanding of scene contents.
> However, our inputs are sparse LiDAR point clouds and structured texts, and we aim to regularize localization.
> We also conduct a new ablation study by replacing LATER with CoOP and Flamingo.
> We retain the original fusion configuration, replacing only the fusion module itself.
> The results shown in the table below indicate that with these fusion mechanisms, we can not achieve optimal results.
> This proves that the unique dynamic refinement mechanism of LATER is both effective and necessary.
>
> |Method|QEOxford|Oxford|NCLT|
> |-|-|-|-|
> |vanilla|0.83/1.12|2.67/1.25|1.46/2.80|
> |CoOP|0.80/1.10|2.62/1.22|1.44/2.70|
> |Flamingo|0.82/1.11|2.65/1.25|1.47/2.79|
> |LATER|0.75/1.03|2.59/1.19|1.40/2.62|
>
> **Response to Q2 Result variability.**
> In our initial submission, we follow the common practice within the LiDAR localization field, where most prior works report mean errors from a single, deterministic run.
> This ensures that our results are directly and fairly comparable to the existing SOTA.
> To further strengthen our conclusions, we have conducted additional experiments to report the variance and confidence interval.
> We re-run our method (GTR-Loc) and the strongest baseline (LightLoc) for 5 iterations using different random seeds on the QEOxford and NCLT datasets.
> The results (mean ± standard deviation) of position/orientation errors are presented in the table below.
>
> |Method|QEOxford|NCLT|
> |-|-|-|
> |LightLoc|0.83 ± 0.012/1.12 ± 0.016|1.46 ± 0.016/2.80 ± 0.018|
> |GTR-Loc|0.75 ± 0.010/1.03 ± 0.014|1.40 ± 0.012/2.62 ± 0.014|
>
> As shown in the table above, the standard deviations across multiple runs are very small, indicating that the performance of both methods is highly stable.
> As shown in the table below, GTR-Loc's variance (Var) of position error and orientation error is consistently and robustly superior to that of LightLoc across all 5 runs.
> The 95\% confidence intervals (CI) of our method's performance do not overlap with the baseline's, confirming that the reported improvements are statistically significant and not an artifact of random chance.
>
> |QEOxford|Var. (Pos. Err.)|Var. (Ori. Err.)|Pos. Err. CI (95\%)|Ori. Err. CI (95\%)|
> |-|-|-|-|-|
> |LightLoc|0.000144|0.000256|0.83 ± 0.0149|1.12 ± 0.0199|
> |GTR-Loc|0.000100|0.000196|0.75 ± 0.0124|1.03 ± 0.0174|
>
> |NCLT|Var. (Pos. Err.)|Var. (Ori. Err.)|Pos. Err. CI (95\%)|Ori. Err. CI (95\%)|
> |-|-|-|-|-|
> |LightLoc|0.000256|0.000324|1.46 ± 0.0199|2.80 ± 0.0224|
> |GTR-Loc|0.000144|0.000196|1.40 ± 0.0149|2.62 ± 0.0174|
>
> **Response to Q3 Adaptivity of discretization.**
> (1) *Learnable discretization schemes.*
> In our current work, we use a fixed, uniform grid partitioning primarily for its interpretability and efficiency.
> A uniform grid is a simple, structure-agnostic approach.
> To explore the potential benefits of adaptive partitioning, we conduct new ablation studies with a learnable strategy.
> Specifically, we use learnable parameters, nn.Parameter, to learn districts z and directions d, with initial z=100 and d=16.
> The results are shown below, where the learnable discretization scheme is not better than ours.
> We think the reason is that a simple learnable scheme may introduce optimization challenges, as this important parameter is difficult to learn.
>
> |Method|QEOxford|Oxford|NCLT|
> |-|-|-|-|
> |Learnable discretization|0.78/1.08|2.64/1.22|1.43/2.72|
> |GTR-Loc|0.75/1.03|2.59/1.19|1.40/2.62|
>
> (2) *Failure cases.*
> First, coarse districts containing geometrically similar but distinct locations (e.g., parallel streets) may receive an identical text signal that fails to provide disambiguation.
> In such cases, performance may revert to the baseline, though this text is benign rather than actively harmful.
> Second, partition boundaries cutting across long roads can cause textual labels to hop during training.
> However, this issue is mitigated by our LATER module, which leverages real-time visual input, and is further smoothed by distillation, ensuring minimal impact on the final inference model.

---

> ### Author Response · Authors · 2025-08-09
> **Thanks for the Constructive Comments**
>
> We thank the reviewers for their constructive and positive comments. We have addressed all points individually and will incorporate all corresponding revisions into the updated manuscript. We hope and have endeavored to resolve all of the reviewers' concerns.

---

### Official Review · Reviewer_kvSg · 2025-07-02

**Clarity:** 2
**Significance:** 3
**Originality:** 3
**Rating:** 4
**Confidence:** 3

**Summary:**

This paper proposes GTR-Loc of the text-assisted localization method for Scene Coordinate Regression (SCR) in pointclouds obtained from LiDAR. Their GTR-Loc incorporates Geospatial Text Generator (GTG) to encode the geospatial text description, such as addresses: "District 99, West-Southwest."  LiDAR-Anchored Text Embedding Refinement (LATER) module generates view-specific text embeddings.  The extracted textual representations are combined with ordinal pointcloud representation by point encoder. They also used Modality Reduction Distillation (MRD) to  enhance LiDAR-only inference from LiDAR-text inference. In experiments, they performed with three different datasets: Oxford, QEOxford and NCLT datasets. They confirmed the effectiveness of their GTG in the ablation study and achieved SoTA performance in QEOxford. Their research idea of injecting geospatial textual knowledge is overall interesting.

**Questions:**

- Q1. Can you clarify how the generated geospatial text description by GTG for each dataset is utilized in the inference? I assume that you use the distilled LiDAR-only inference module in the test set. If you use textual description in both the training and inference, please clarify how did you prepare geospatial textual description for test sets of those datasets. Also see W4(1) and (2).
- Q2. How rich is the Geospatial text description? Is it merely the combination of positional and

**Ethical Concerns:**

["NO or VERY MINOR ethics concerns only"]

**Final Justification:**

The use of the text is limited to the training, while it is used for the regularization in the inference: this result sounds a bit strange, although it doesn't deviate from the task setting nor fair comparisons with other studies. I do not strongly recommend the acceptance of the paper although I cannot find critical faults that prevent the acceptance of this paper.

**Limitations:**

Overall achievement looks good, although I am not sure if this methodology works in the real scenes with rich contexts considering the limited textual diversity.

**Quality:**

3

**Strengths And Weaknesses:**

Strength
- [S1] The research idea novelty: the generation of geospatial textual description is interesting and indeed the effectiveness of the proposed method is (at least partially) confirmed in the experiments.
- [S2] The effectiveness of the GTG is confirmed by the ablation study, which supports the main contribution of the generating and utilizing geospatial textual description.
- [S3] The SoTA performance in the QEOxford dataset.


Weakness
- [W1] The lack of the diversity in the geospatial textual descriptions: they seemed written following a very limited template. For example, although Fig.3 presents the “free-form scene description,” they used template-based Geospatial text description that is a combination of positional and orientational labels, not rich textual expressions.
- [W2] As I explained in W1, GTG depends on some templates and I suspect that it is focused on the specific environments, e.g., those in the QEOxford dataset.
- [W3] The analyses why the proposed model performed well on the QEOxford dataset while it performed on-per with their baselines in other two datasets are not well-discussed.
- [W4] Writing issue: The authors can include further details of the proposed models, e.g., (1) how they prepare geospatial textual descriptions in the inference time if they are needed, (2) whether they use some additional information (texts) that are not used in the previous studies in the inference time or not.

---

> ### Author Rebuttal · Authors · 2025-07-30
>
> Thank you for your insightful comments.
> We have addressed each point and will integrate all corresponding revisions into the final version of the paper.
>
> **Response to W1\&Q2.**
> (1) *Disadvantages of rich text for LiDAR localization.*
> Human-like text generation, such as TOD3Cap in Figure 3, tends to describe target-environment-relationship, such as vehicles or vegetation.
> These free-form scene descriptions are inherently unstable; dynamic objects and environmental changes lead to inconsistent characterizations of the same location over time, providing an unreliable description.
> Furthermore, these descriptions offer only indirect cues for 6-DoF pose estimation, with irrelevant details acting as noise that can impede learning.
> This challenge is compounded by the sparse, geometric-only nature of LiDAR point clouds, which, unlike images, lack rich texture and color.
>
> (2) *Advantages of GTG for LiDAR localization.*
> In contrast, our GTG produces text (e.g., "District 99, West-Southwest") that yields a standardized representation of universal pose information, creating a signal that is stable, repeatable, and directly relevant to pose estimation.
> The text's invariance to dynamic content provides the consistency needed to resolve localization ambiguities via regularization.
> Then, the LATER module refines GTG-generated text by fusing it with the current point cloud's geometry, creating viewpoint-specific embeddings that enrich the final training feature with diverse visual-geometric information beyond the core pose.
>
> (3) *Experimental validation and proof.*
> In Table 4, we demonstrate that GTG is more effective than free-form scene descriptions.
> We also present this table here.
> We replace GTG with a SOTA 3D scene captioning model, TOD3Cap, to generate rich text.
> This free-form text mainly describes object motion, appearance, relationship, and environment.
> However, using free-form descriptions from TOD3Cap yields no significant performance gain over the vanilla model.
> Our template-based GTG leads to significant improvements.
>
> |Method|QEOxford|Oxford|NCLT|
> |-|-|-|-|
> |vanilla|0.83/1.12|2.67/1.25|1.46/2.80|
> |TOD3Cap|0.84/1.15|2.66/1.24|1.47/2.78|
> |GTR-Loc|0.75/1.03|2.59/1.19|1.40/2.62|
>
> **Response to W2.**
> *The proposed GTG does not focus on specific environments.*
> First, as discussed in the previous response, the GTG is not tied to any specific environmental features like roads or buildings.
> It partitions any given map into discrete districts and directions, which is a universal geospatial cue that can be applied to any environment that can be mapped.
> Second, the proposed GTG and the entire text modality are used exclusively during training and are completely absent during inference.
> We propose MRD distillation to conduct LiDAR-only localization at inference, without requiring text inputs.
> A localization model that does not use text at inference avoids overfitting to the templates of any specific environment.
>
> **Response to W3.**
> *Performance analysis.*
> We would like to respectfully clarify that GTR-Loc achieves superior performance on all three datasets.
> Our method outperforms all baselines on QEOxford, achieves SOTA position accuracy on Oxford and NCLT, and ranks second in orientation estimation.
> Specifically, on QEOxford, GTR-Loc (0.75m/1.03°) achieves a significant 9.64\%/8.04\% improvement in both position and orientation over the previous SOTA, LightLoc (0.83m/1.12°).
> GTR-Loc consistently outperforms LightLoc on both the Oxford (2.59m/1.19° vs. 2.67m/1.25°) and NCLT (1.40m/2.62° vs. 1.46m/2.80°) datasets.
> The absolute improvement in position accuracy over previous leading SCR methods, LiSA and LightLoc, is significant.
>
> The primary goal of GTR-Loc is to solve localization ambiguity, where a model incorrectly identifies its location due to geometrically similar scenes.
> The Oxford dataset, including QEOxford (a version with enhanced ground-truth), features long, repetitive urban roads with highly similar geometric structures, which creates significant ambiguities.
> Hence, the margin of improvement is most pronounced on QEOxford and still significant on Oxford.
> The NCLT dataset, gathered via a Segway, is less stable than data from vehicle platforms, making localization more challenging.
> Nevertheless, our superior performance against the SOTA method, LightLoc, validates the effectiveness of using geospatial text regularization.
>
> **Response to W4\&Q1.**
> (1) *If inference with text.*
> It is crucial to distinguish our final model—a distilled, LiDAR-only regressor (red flow shown in Figure 2) requiring only a single LiDAR scan for a fair comparison—requires no text at inference.
> To address the reviewer's concern and strengthen our results, we have conducted additional experiments using GTR-Loc with text during inference.
> Specifically, we first train an auxiliary classification network to predict district and direction categories from LiDAR point clouds.
> We set this network's architecture as a LightLoc encoder followed by four FC layers.
> During inference, the pre-trained classification network predicts the district and direction ID for text generation.
> This generated text, along with the original point cloud, is then fed into the LiDAR-text Regressor (blue flow shown in Figure 2) to perform prediction.
>
> |Method|QEOxford|Oxford|NCLT|
> |-|-|-|-|
> |vanilla|0.83/1.12|2.67/1.25|1.46/2.80|
> |GTR-Loc w text at inference|0.75/1.02|2.60/1.21|1.41/2.70|
> |GTR-Loc|0.75/1.03|2.59/1.19|1.40/2.62|
>
> The results indicate that this approach is not effective.
> It delivers virtually no performance gains (the table above) while increasing both test time and computational complexity due to the additional classification network and multimodal regression.
> Inaccurate classification (the table below) leads to wrong text generation, thus resulting in inaccurate localization.
> Therefore, it further demonstrates that transferring knowledge via MRD into a LiDAR-only model is more efficient and effective.
> These additional experiments have also been reported in Tables 3 and 4 of the appendix.
>
> |Dataset|Accuracy|
> |-|-|
> |QEOxford|99.19\%/97.64\%|
> |Oxford|98.99\%/97.52\%|
> |NCLT|98.44\%/85.50\%|
>
> (2) *Additional information for inference.*
> There is no additional text information used for GTR-Loc during inference.
> Our model's input is identical to that of previous SCR methods like LightLoc—only the LiDAR point cloud.
> The proposed MRD distillation allows LIDAR-only inference.
> Once training is complete, this model has no access to GTG, text, or pose information.

---

> > ### Comment · Reviewer_kvSg · 2025-08-06
> > **Thank you for clarification**
> >
> > The AR is helpful and it seems that no text is required at inference although it is needed in the training. Using text in the inference doesn't have much impact on the score according to Response to W4&Q1. Although this indeed sounds a bit strange, it doesn't affect the main state of the paper as the regularizer effect of the text in the proposed Geospatial Text Regularization methodology.

---

> ### Author Response · Authors · 2025-08-07
> **Response Update to W4&Q1**
>
> Thank you for your constructive feedback. We have addressed W4&Q1 by including additional experimental setups and results. We welcome any further discussion and deeply appreciate your guidance in improving this paper.
>
> Using geospatial text for inference creates a circular dependency, as its construction requires the very pose information we aim to estimate. Therefore, we propose MRD to perform LiDAR-only localization during inference. Nevertheless, we evaluate its potential by experimenting with text generated from (1) predicted and (2) gt pose-relevant information.
>
> (1) As explained for W4&Q1, we generate text by using a classification network to predict pose-relevant categories (district/direction). However, even with the GTG template, errors from this classification (row 5) lead to flawed text that hinders localization. Consequently, it yields no discernible performance gain (row 2, consistent with our W4&Q1 response).
>
> (2) We then test a hypothetical scenario where gt poses are available for text generation at inference. This better result (row 3) establishes the theoretical upper bound for performance achievable with text. However, it is impractical, as gt poses are unavailable during real-world inference. Therefore, using MRD distillation is a more viable solution.
>
> ||Method|QEOxford|Oxford|NCLT|
> |-|-|-|-|-|
> |1|vanilla|0.83/1.12|2.67/1.25|1.46/2.80|
> |2|w pred text|0.75/1.02|2.60/1.21|1.41/2.70|
> |3|w gt text|0.74/0.93|2.58/1.07|1.12/2.40|
> |4|GTR-Loc|0.75/1.03|2.59/1.19|1.40/2.62|
> |5|Classification Accuracy|99.19%/97.64%|98.99%/97.52%|98.44%/85.50%|

---

> > ### Comment · Reviewer_kvSg · 2025-08-09
> > **Thank you**
> >
> > Thank you. As sufficient information seems gathered, I finalize my decision through the later reviewer discussion phrase.

---

> > > ### Author Response · Authors · 2025-08-09
> > > **Acknowledgment to the reviewer**
> > >
> > > Thank you for your time and for the thorough review. We appreciate your insightful feedback and are glad that our responses were helpful. We look forward to the final decision.

---

### Note · Authors · 2025-08-13

We thank the reviewers for their constructive feedback and for recognizing our work’s **clear motivation (ispn, zga8, YNwr)**, **strong results (kvSg, ispn, YNwr)**, and **high novelty (kvSg, zga8, YNwr)**. These discussions have substantially improved the paper's arguments and technical rigor.

**Contributions & Novelties of Our Work**

1. We introduce GTR-Loc, a new method that uses geospatial text as regularization to resolve LiDAR localization ambiguities where traditional geometric cues fail.

2. We propose two novel modules: the Geospatial Text Generator (GTG) for creating structured texts, and the LiDAR-Anchored Text Embedding Refinement (LATER) module for dynamic, view-specific refinement.

3. We design the Modality Reduction Distillation (MRD) that transfers text-assisted knowledge into a LiDAR-only model, removing the need for text at inference.

**Major Improvements During Rebuttal**

1. "Simplicity" of Text: Our structured, pose-aware text is more effective than semantic scene descriptions; the LATER module adds the necessary dynamic nuance.

2. Use of Text at Inference: The final model is LiDAR-only. Generating text at inference is ineffective, validating our MRD distillation.

3. Text's Effectiveness: Text regularization resolves core ambiguities rather than just accelerating training.

4. CLIP's Necessity: CLIP provides superior spatial understanding and performance compared to an embedding layer trained from scratch.

5. Partitioning Robustness: Our partitioning strategy is robust to the granularity of districts and directions.

6. Statistical Significance: Multi-seed experiments with confidence intervals confirm our gains are statistically significant.

7. Efficiency: GTR-Loc achieves real-time inference speeds, with a modest training overhead.

**Takeaway:** We believe GTR-Loc offers a significant impact by proposing geospatial text regularization to solve a key bottleneck in LiDAR localization. We will integrate all clarifications and release our code upon acceptance.

We sincerely thank the AC and SAC for their time throughout this process.

---

### Decision · Program_Chairs · 2025-09-17

**Decision:**

Accept (poster)

**Comment:**

In the paper the author introduce a method called GTR-Loc, which use a combination of text cues and LIDAR data. The main contribution is this novel combination framework, i.e. the use of textual descriptions (here it was decided to use fairly specific type of text cues that give rich geometric information) to mitigate localization ambiguities in large-scale outdoor environments. To handle intra-region variability, the authors propose a LiDAR-Anchored Text Embedding Refinement (LATER) module that adapts text embeddings based on local point cloud features. The model is trained using a dual-regression architecture with a Modality Reduction Distillation (MRD) strategy to enable efficient LiDAR-only inference. The authors conduct experiments showing that the proposed method consistently improves localization robustness. Furthermore, the paper presents extensive ablation analyses, demonstrating that the proposed components contribute meaningfully to resolving ambiguity in visually similar scenes.
Strengths include (i) novel ideas, (ii) solid ablation study, (iii) solid experimental validation, which show state of the art performance on the QEOxford dataset, (iv) well designed (modular) architecture that allows for improvement (and ablation studies).
Weaknesses include (i) it seems that the method was designed with the geometrically informative text descriptions in The QEOxford dataset and results were only on par with other methods for the other datasets, (ii) discussions could be improved, e g on the use of CLIP for such limited type of text descriptions and on the ablation studies.
The authors and reviewers were actively engaged in discussion during the rebuttal period and the authors made a good job in answering weaknesses/questions posed by the reviewers. In the end reviewers were all for accepting the paper and I agree and therefore recommend accepting the paper.